# Implications of sea-ice biogeochemistry for oceanic production and emissions of dimethylsulfide in the Arctic

Hakase Hayashida[1], Nadja Steiner[2], Adam Monahan[1], Virginie Galindo[3], Martine Lizotte[4], and Maurice Levasseur[4]

[1]School of Earth and Ocean Sciences, University of Victoria, Victoria, British Columbia, Canada
[2]Institute of Ocean Sciences, Fisheries and Oceans Canada, Sidney, British Columbia, Canada
[3]Centre for Earth Observation Science, Faculty of Environment, Earth and Resources, University of Manitoba, Winnipeg, Manitoba, Canada
[4]Département de biologie, Québec-Océan, Université Laval, Québec, Québec, Canada

*Correspondence to:* Hakase Hayashida (hakase@uvic.ca)

**Abstract.** Sea ice represents an additional oceanic source of the climatically active gas dimethylsulfide (DMS) for the Arctic atmosphere. To what extent this source contributes to the dynamics of summertime Arctic clouds is however not known due to scarcity of field measurements. In this study, we developed a coupled sea ice-ocean ecosystem-sulfur cycle model to investigate the potential impact of bottom-ice DMS and its precursor dimethylsulfoniopropionate (DMSP) on the oceanic production and emissions of DMS in the Arctic. The results of the 1-D model simulation were compared with field data collected during May and June of 2010 in Resolute Passage. Our results reproduced the accumulation of DMS and DMSP in the bottom ice during the development of an ice algal bloom. The release of these sulfur species took place predominantly during the earlier phase of the melt period, resulting in an increase of DMS and DMSP in the underlying water column prior to the onset of an under-ice phytoplankton bloom. Production and removal rates of processes considered in the model are analyzed to identify the processes dominating the budgets of DMS and DMSP both in the bottom ice and the underlying water column. When openings in the ice were taken into account, the simulated sea-air DMS flux during the melt period was dominated by episodic spikes of up to 8.1 $\mu$mol m$^{-2}$ d$^{-1}$. Further model simulations were conducted to assess the effects of the incorporation of sea-ice biogeochemistry on DMS production and emissions, as well as the sensitivity of our results to changes of uncertain model parameters of the sea-ice sulfur cycle. The results highlight the importance of taking into account both the sea-ice sulfur cycle and ecosystem in the flux estimates of oceanic DMS near the ice margins and identify key uncertainties in processes and rates that should be better constrained by new observations.

## 1 Introduction

Dimethylsulfide (DMS) is a volatile biogenic compound that is produced primarily through ecological interactions encompassing marine microbial food webs (Simo, 2001). Oceanic emissions of DMS are the largest natural source of sulfur in the atmosphere (Bates et al., 2004), thereby playing a crucial role in global sulfur cycling. Oceanic DMS emissions can play an important role in climate because oxidation products of DMS can serve as atmospheric aerosols and cloud condensation nuclei

(CCN), therefore contributing to radiative forcing (Shaw, 1983). In 1987, Charlson et al. hypothesized that enhanced oceanic DMS emissions due to global warming could produce a negative feedback via increased scattering of incoming shortwave radiation by DMS-derived aerosols and CCN. Although this climate regulation by oceanic DMS emissions has been suggested to be unlikely on a global scale (Quinn and Bates, 2011), oceanic DMS emissions could still exert a significant influence on local climate in certain regions, such as the Arctic (Chang et al., 2011; Levasseur, 2013).

During the mid-spring and summer (May-August), the Arctic atmosphere becomes relatively free of anthropogenic aerosols due to increased wet deposition and decreased transport from lower latitudes (Croft et al., 2016). At the same time, concentrations of methanesulfonic acid (MSA), an oxidation product of DMS, have been observed to increase and peak at various locations north of 70° N (Sharma et al., 2012; Willis et al., 2016). The cleansing of the summertime Arctic atmosphere and the emergence of relatively high concentrations of MSA point towards oceanic DMS as the driver for the formation and growth of new particles (Sharma et al., 2012; Leaitch et al., 2013), along with other important biogenic sources of CCN, such as microgels (Orellana et al., 2011; Tjernström et al., 2014). Simultaneous measurements of sea surface and atmospheric DMS concentrations provide further evidence linking new particle formation events to oceanic DMS emissions (Chang et al., 2011; Rempillo et al., 2011).

In addition to DMS produced within the water column, the presence of sea ice provides an additional source of oceanic DMS in the Arctic that can make a transient but potentially important contribution to the formation of sulfur-containing aerosols and clouds during the melt period (Levasseur et al., 1994; Levasseur, 2013; Mungall et al., 2016). Especially during spring (April-June), DMS and its precursor dimethylsulfoniopropionate (DMSP) can reach very high concentrations in the bottom layer of Arctic sea ice throughout the development of the ice algal bloom (Levasseur et al., 1994). Measurements of DMS and DMSP reveal concentrations in the bottom ice that are often 1 to 3 orders of magnitude larger than in the under-ice and open water columns (Levasseur et al., 1994; Uzuka, 2003; Levasseur, 2013; Galindo et al., 2014, 2015). How much of this ice-related DMS eventually reaches the atmosphere is not known, but mechanisms have been suggested by which the DMS produced in the bottom ice supplies pulses of DMS into the pristine Arctic atmosphere during spring and therefore contributes significantly to the formation of new clouds in the Arctic (Levasseur et al., 1994). However, it is difficult in practice to measure the sea-air flux of DMS originating from the bottom ice alone and therefore to quantify the contribution of that flux relative to DMS produced within the water column. Process models can aid the understanding of the relevance of specific processes to the Arctic marine sulfur cycle as well as their likely spatio-temporal variability. To the best of our knowledge, only one previous study has incorporated the sea-ice sulfur cycle in model simulations (Elliott et al., 2012). This earlier study demonstrated that the DMS production in the bottom ice can supply a significant amount of DMS (exceeding 10 nmol L$^{-1}$) in the upper mixed layer in many locations in the Arctic Ocean. However, neither the importance of bottom-ice source relative to the production in the underlying water column was assessed nor was an attempt made to provide the potential emissions into the atmosphere in this previous study.

In the present study, we test the hypothesis that DMS and DMSP produced in the bottom ice can make a substantial contribution to the production and emissions of oceanic DMS in the Arctic by developing a sulfur cycle module for the bottom ice

and underlying water column. This module was embedded into a coupled sea ice-ocean ecosystem model to conduct various simulations which were compared to observations within landfast first-year ice in Resolute Passage during 2010.

## 2 Model description and experimental design

A sulfur cycle module for the bottom ice and the water column was developed and embedded into an existing coupled sea ice-ocean ecosystem model. The resulting coupled model was applied in a one-dimensional (1-D) configuration to conduct simulations of DMS and DMSP dynamics within and under the bottom layer of landfast first-year ice in Resolute Passage during 2010.

### 2.1 Ecosystem model

The coupled sea ice-ocean ecosystem model is described and evaluated in Mortenson et al. (2016). In this earlier study, the model was used to study the physical and biological controls on the ice algal and under-ice phytoplankton blooms observed in Resolute Passage during the spring of 2010. The sea-ice component of the model is based on Lavoie et al. (2005) and consists of four prognostic variables including nitrate, ammonium, silicate, and ice algae. The model simulates the growth and decline of ice algae in the bottom layer of the sea ice, as well as the release of ice algae into the water column during the melt period. The oceanic component of the model is a ten-compartment (nitrate, ammonium, silicate, small and large phytoplankton, small and large zooplankton, small and large detritus, and particulate silica) lower-trophic level ecosystem model derived from Steiner et al. (2006). In the uppermost layer of the water column, the ocean ecosystem model is coupled to the sea-ice ecosystem model to represent the diffusive exchange of nutrients at the ice-water interface, as well as the release of living and dead ice algae into the water column as large phytoplankton and large detritus, respectively. The ecosystem dynamics are driven by physical processes which are computed by a coupled sea ice-ocean physical model. The oceanic component of this model is the General Ocean Turbulence Model (GOTM), a public domain 1-D water column model (Burchard et al., 1999, 2006). Horizontal velocity fields, turbulent transports, photosynthetically active radiation (PAR), temperature, and salinity in the water column are simulated by GOTM and are provided for simulation of pelagic ecosystem and sulfur cycle dynamics. The sea-ice component is based on the 1-D thermodynamic model developed by Flato and Brown (1996), which consists a single layer of snow and multi-layers of ice. In the present version, the model considers non-uniform snow thickness distributions and melt ponds, which improved the simulation of light fields under snow and sea ice (Abraham et al., 2015). Ice growth/melting rate, melt-pond areal fraction, and basal ice temperature are simulated by the sea ice model and are provided for simulation of bottom-ice ecosystem and sulfur cycle dynamics.

### 2.2 Sulfur cycle module

The ocean can be seen as an infinite reservoir of sulfur for the atmosphere, although its contribution to the atmospheric sulfur budget depends on how much of this reservoir can be converted into the volatile compound DMS. The structural design of the coupled sea ice-ocean sulfur cycle module developed in the present study was inspired mainly by two previous marine sulfur

cycle models (Archer et al., 2004; Steiner and Denman, 2008). It should be emphasized that the sulfur cycle represented in this and earlier studies considers the cycling between DMSP and DMS only, and does not conserve total sulfur. However, total sulfur conservation is not a requirement because sulfur is not a limiting nutrient for primary producers and sea-air fluxes only depend on the concentration of DMS. Figure 1 shows the variables and processes represented in the module that are deemed

most relevant for the production and removal of DMSP and DMS in the bottom ice and water column. DMSP in particulate (DMSPp) and dissolved (DMSPd) phases are simulated separately as they have distinct physical properties and ecological roles in sulfur cycling. For example, DMSPp released from the bottom ice is expected to sink quickly through the water column, whereas DMSPd likely remains in the under-ice meltwater lens upon its release from the bottom ice (Elliott et al., 2012; Galindo et al., 2014, 2015). Furthermore, only DMSPd can be assimilated by bacteria to produce DMS (Stefels et al.,

2007). In the model, DMSPp is simulated diagnostically by assuming a fixed intracellular DMSP:Chlorophyll $a$ (Chl $a$) ratio for each of the simulated algal groups, while DMSPd and DMS are simulated prognostically. DMSPd is produced by cell lysis and exudation, and in the case of the water column, by sloppy feeding, while it is removed by bacterial consumption and free DMSP-lyase. DMS is produced by bacterial DMSPd-to-DMS conversion and free DMSP-lyase, while it is removed by bacterial consumption, photolysis, and in open water conditions, by sea-to-air flux. Due to the absence of rate measurements within sea

ice, most of the parameters prescribed for these simulated processes are taken from limited water column measurements (see Appendix A for details). In the water column, all sulfur species are mixed between model layers with eddy diffusivities computed by the ocean physical model. At the ice-water interface, the sulfur species are coupled one way through the release of DMSP-containing ice algae, DMSPd, and DMS from the bottom ice into the uppermost layer of the water column. The concentrations of simulated sulfur species are computed at each model layer by a system of differential equations representing

the budgets of these species, with parameterized expressions for the processes discussed above. A detailed description of the sulfur cycle module is presented in Appendix A. A detailed discussion of neglected physical and biogeochemical processes that may potentially be important to the sulfur cycle is presented in Section 3.3.

## 2.3  Study site

The focus of this study is landfast first-year ice in Resolute Passage, Nunavut, Canada. This site was chosen because of the

availability of extensive datasets from previous field studies on ice-associated ecosystems and biogeochemistry (e.g. Smith, 1988; Levasseur et al., 1994; Lavoie et al., 2005; Michel et al., 2006; Mundy et al., 2014), including time series of sea-ice DMSP measurements (Galindo et al., 2014, see below). Furthermore, situated in the central Canadian Arctic Archipelago, Resolute Passage is representative of Arctic continental shelves which constitute more than 50 % of the total area of the Arctic Ocean (Jakobsson et al., 2003) and represent more than 80 % of the total primary production in the high Arctic (Sakshaug,

2004). The landfast first-year ice found in Resolute Passage can reach a thickness of more than 2 m (e.g. Flato and Brown, 1996) and typically remains in the region until July (e.g. Galindo et al., 2014).

During May and June of 2010, continuous measurements of DMSPp and DMSPd within and under the sea ice in Resolute Passage were carried out as part of a time-series ice study called the Arctic Ice-Covered-Ecosystem (Arctic-ICE) project (Galindo et al., 2014). As the sampling was done from a single sea ice, it was deemed ideal to test our 1-D model for this Arctic-ICE

2010 study. Data from this field campaign were used to calibrate the parameters of the sulfur cycle module and evaluate the results of model simulations.

## 2.4 Model setup

The model developed in this study was applied to the study site of the Arctic-ICE 2010 field campaign (74°42.6' N and 95°15' W; Galindo et al., 2014). The vertical domain of the model was divided into 10 uniformly spaced layers for the sea ice and 100 uniformly spaced layers for the upper 100 m of the water column (the actual depth of the water column of the study site was 141 m; Galindo et al., 2014). The model was integrated with a time step of 10 minutes from 1 February to 6 July, 2010. At the surface, the model was forced with Environment Canada's hourly weather data (including surface 2-m air temperature, zonal and meridional wind at 10 m above the sea surface, surface air pressure, relative humidity, cloud cover, and precipitation; http://climate.weather.gc.ca/) collected at Resolute airport, which is located within 7 km of the study site. A meteorological station deployed at the study site only provided limited time coverage (i.e. between May and June) and a limited set of variables (i.e. air temperature and irradiance), however, surface 2-m air temperature profile measured at the airport (Fig. S1) compares well with the time series collected at the meteorological station (Fig. 2a of Mundy et al., 2014). Simulated temperature, salinity, and horizontal velocity fields were restored over the entire water column to the output of a simulation from a coupled 3-D regional sea ice-ocean circulation model (NEMO-LIM2; Dukhovskoy et al., 2016, and references therein) with restoring timescales of 1 day for temperature and salinity, and 10 minutes for horizontal velocity fields. Initial snow and melt pond depths and ice thickness set respectively to 5, 0, and 55 cm, result in simulations of these variables in good agreement with the measurements from the Arctic-ICE 2010 field campaign. Similarly, to simulate an ice algae bloom comparable to that in the the Arctic-ICE 2010 study, the initial biomass of ice algae was set to 3.5 $\mu$g Chl $a$ L$^{-1}$. Although this value may seem high, previous studies report a wide range of Chl $a$ concentrations in young sea ice (0.3-26.8 $\mu$g Chl $a$ L$^{-1}$) that is often higher than concentrations in the water column (e.g. Garrison et al., 1983). The thickness of the bottom-ice skeletal layer (in which the ecosystem and sulfur processes take place) was set to 3 cm and matches the vertical sampling resolution of Galindo et al. (2014). The initial concentrations of nitrate and silicate in the bottom ice and water column were respectively set to 7.2 $\mu$mol N L$^{-1}$ and 14.7 $\mu$mol Si L$^{-1}$, based on measurements at the beginning of the Arctic-ICE 2010 field campaign (Mundy et al., 2014). The initial concentrations of ammonium in the bottom ice and water column, as well as the remaining ocean ecosystem model variables were set to 0.01 $\mu$mol N L$^{-1}$ ($\mu$mol Si L$^{-1}$ for particulate silica). The initial concentrations of DMSPd and DMS were assumed to be small, and were set to 0.1 nmol S L$^{-1}$ in the bottom ice and water column.

## 2.5 Model experiments

Two types of model simulations were conducted in this study: standard and sensitivity runs. The standard run was designed to simulate the observed variability of physical and biogeochemical variables during the Arctic-ICE 2010 field campaign. Specifically, the performance of the standard run was evaluated by directly comparing the simulated results with the observed time series of snow and melt pond depths, ice thickness, Chl $a$, DMSPp and DMSPd in the bottom ice and upper water column.

The default values of the sulfur cycle model parameters (Table 1) were calibrated to match the observations, starting from initial guesses based on both previous model studies and available field measurements in Arctic waters (see Appendix).

Three types of sensitivity runs were designed to assess the impact of sea-ice biogeochemistry on the production and emissions of DMS under the ice. The first experiment evaluated the changes in the simulated under-ice DMSPd and DMS concentrations due to the presence or absence of sea-ice biogeochemistry. The second experiment explored the model uncertainty resulting from uncertainties in the parameters of the sea-ice sulfur cycle. The third experiment quantified the potential sea-air fluxes of DMS through openings in the ice during the melt period and the relative contributions of the sea-ice sulfur cycle and ecosystem to those fluxes. Details of the sensitivity runs are described in Section 3.2.

## 3 Results and discussions

### 3.1 Standard run

#### 3.1.1 Snow, melt ponds, and sea ice

Figure 2a shows the simulated and observed time series of snow and melt pond depths and ice thickness. It is important to note that our model accounts for subgrid-scale snow depth variability (Abraham et al., 2015), hence the simulated results are intended to represent an areal average over the study site (as would be the case for an individual grid cell in a global or regional model). Because the observations were taken at multiple locations with different snow depths on most days, the mean of these observations can be directly compared with the simulated results. During the winter and spring, the simulated snow depth increased as a result of occasional snowfall events until it reached about 20 cm in mid-May (black solid line; Fig. 2a). Simulated snow started melting at the end of May, and had disappeared completely by mid-June. The simulated snow depth is close to the observed site-average snow depth (black dots; Fig. 2a). The resulting melt water from the simulated snow contributed to the formation of simulated melt ponds that reached a mean depth of about 2 cm in late June (black dashed line; Fig. 2a). The timing of simulated melt pond formation is reasonable as melt ponds with similar depths were observed during the last two days of sampling, as indicated by the negative values in the observed snow depth range. The simulated ice thickness increased gradually until it reached about 145 cm in early June (red line; Fig. 2a). Simulated ice melt started shortly after the initiation of snowmelt and was complete by early July. The observed range of ice thickness was small indicating its homogeneity over the study site (red vertical bars; Fig. 2a), and is comparable to the simulated values. Furthermore, the timing of the simulated ice disappearance is close to the timing of the ice breakup observed in the field (mid-July; Galindo et al., 2014). This indicates a dominance of thermodynamic processes which is expected for the region (Flato and Brown, 1996).

#### 3.1.2 Ice algae and phytoplankton

Figure 2b shows the simulated and observed time series of ice algal biomass in the bottom 3 cm of the sea ice and phytoplankton biomass averaged over the upper 10 m of the water column. The simulated ice algal biomass increased gradually from late March and reached about 1100 $\mu$g Chl $a$ L$^{-1}$ by mid-May (black line; Fig. 2b). The simulated ice algal biomass did not increase

further due to nitrogen limitation during the remainder of May and decreased rapidly due to release into the underlying water column associated with both flushing and basal melting during the melt period in June (Mortenson et al., 2016). The simulated ice algal bloom terminated in late June, about two weeks prior to the simulated ice breakup (red line; Fig. 2a). Both the magnitude and temporal variations in the simulated ice algal bloom are generally comparable with the observations at the study site (black dots; Fig. 2a).

In the upper 10 m of the water column, the simulated phytoplankton biomass started increasing in early June and quickly reached a peak of about 11 $\mu$g Chl $a$ L$^{-1}$ in mid-June (red line; Fig. 2b). This simulated under-ice phytoplankton bloom was dominated by large cells and terminated due to nitrogen limitation (Mortenson et al., 2016). These findings are consistent with the observations that reported that the bloom was numerically dominated by centric diatoms and led to the complete use of nitrate and nitrite (down to about 0.1 $\mu$mol L$^{-1}$) in the upper 10 m of the water column (Mundy et al., 2014). The timing and magnitude of the simulated under-ice phytoplankton bloom are generally comparable with the observations, except for those increases in the observed phytoplankton biomass during the first few sampling days in early May and during four sampling days in early June (red dots; Fig. 2b). Based on the bulk salinity measurements, Galindo et al. (2014) concluded that brine drainage was occurring prior to the snowmelt period. Therefore, the first model-observation mismatch in May was likely due to the release of ice algae by brine drainage, which was not simulated by our model. In contrast, the second mismatch in early June can not be explained by brine drainage, as the observed bulk salinity was fairly constant during this period (Galindo et al., 2014). Since this mismatch occurred during the snowmelt period, we hypothesize that the model might have underestimated the release due to flushing.

### 3.1.3 DMSPp, DMSPd, and DMS concentrations

Figure 3a shows the simulated and observed time series of the bottom-ice (3 cm) DMSPp concentration and the seawater DMSPp concentration averaged over the upper 10 m of the water column. The simulated DMSPp concentrations were determined by assuming fixed DMSPp-to-Chl $a$ ratios, as the observations in Resolute Passage showed a strong linear relationship between DMSPp and Chl $a$ concentrations both in the bottom ice and in the underlying water column ($r^2 = 0.9$; Galindo et al., 2014). In the standard run, these ratios were set to 9.5 nmol S:mg Chl $a$ for ice algae and large phytoplankton, while the ratio of 100 nmol S:mg Chl $a$ was prescribed for small phytoplankton (see Appendix A). Therefore, the temporal variability in simulated DMSPp in the bottom ice was identical to that of simulated ice algal biomass (black line; Fig. 2b). Similarly, in the absence of small phytoplankton, the temporal pattern of simulated DMSPp in the underlying water column follows closely that of simulated phytoplankton biomass dominated by large cells (red line; Fig. 2b). The simulated bottom-ice DMSPp concentration reached about $10^4$ nmol L$^{-1}$ at the peak of the simulated ice algal bloom in mid-May (black line; Fig. 3a). The observed bottom-ice DMSPp concentrations were highly variable during this period both spatially (vertical bars associated with black dots on each sampling day; Fig. 3a) and temporally (the range of black dots; Fig. 3a). The spatial variability likely reflects the patchiness of ice algae collected over sites of various snow cover, while the temporal variability can be related to various stages of the ice algal bloom combined with the effect of brine drainage as discussed in the previous section. The simulated bottom-ice DMSPp concentrations were close to the site-average value observed on day 2, near the lower end of the observed range on day 3, and

close to the upper ends of the observed range on days 4, 6, and 7 of the sampling during May (black dots and associated vertical lines; Fig. 3a). During the melt period in June, the temporal variations in simulated bottom-ice DMSPp concentration closely followed the observed site-average values sampled on the last four days (black dots; Fig. 3a).

In the upper 10 m of the water column, the simulated seawater DMSPp concentration started increasing in June and peaked
at about 100 nmol L$^{-1}$ in mid-June (red line; Fig. 3a), coinciding with the simulated under-ice phytoplankton bloom (red line; Fig. 2b). The simulated values were close to the observed values throughout the sampling period except for days 2-3 and 9-12 (red dots; Fig. 3a). These mismatches are consistent with those found in the under-ice phytoplankton biomass time series, suggesting that they could be attributed to the lack of brine drainage effect (days 2-3) and a potentially underestimated effect of flushing in the model (days 9-12).

Figure 3b shows the simulated and observed time series of the bottom-ice DMSPd concentration and the seawater DMSPd concentration averaged over the upper 10 m of the water column. The simulated bottom-ice DMSPd concentrations gradually increased from early April to late May with a peak of about 1800 nmol L$^{-1}$ (black line; Fig. 3b). During the melt period, the simulated bottom-ice DMSPd concentration decreased gradually and reached nearly 0 nmol L$^{-1}$ by late June. The simulated bottom-ice DMSPd closely followed the observed site-average values except for high values (ca. 5000 nmol L$^{-1}$) measured
during the first three sampling days (black dots; Fig. 3b). Considering that brine drainage had occurred during these sampling days, it may have promoted the production of DMSPp and its conversion to DMSPd (and DMS) due to stress, which is not represented adequately in the model.

In the upper 10 m of the water column, the simulated DMSPd concentrations were nearly 0 nmol L$^{-1}$ until the onset of the simulated under-ice bloom in June (red line; Fig. 2b). In contrast, the observed site-average DMSPd concentrations were above
1 nmol L$^{-1}$ for four consecutive sampling days (from day 3 to 6) in May (red dots; Fig. 3b). This observed DMSPd increase prior to the melt period is consistent with the observed increases in under-ice Chl $a$ and DMSPp, suggesting the influence of brine drainage (red dots on days 2 and 3; Fig. 3a). These increases in observed Chl $a$, DMSPp, and DMSPd in the upper 10 m of the water column gradually ceased and reached nearly nil by the following sampling days (day 4 for DMSPp, day 5 for Chl $a$, and day 7 for DMSPd), which might be explained by a combination of the following two processes. First, some of ice
algal cells and DMSPp released through brine drainage sank quickly into the water column. This argument is supported by a slight increase in observed Chl $a$ and DMSPp at depths below 10 m on the following sampling days, and more prominently, by a larger increase in DMSPd at 50 m depth (ca. 1 nmol L$^{-1}$), which might suggest a degradation of DMSPp-possessing sinking ice algal cells (Figures 8 of Galindo et al., 2014). The other process contributing to the decreases in Chl $a$, DMSPp, and DMSPd in the upper 10 m of the water column could be the degradation of ice algal cells in the upper layer, which could
explain the delay in the decrease in observed DMSPd relative to the decreases in Chl $a$ and DMSPp.

In late June, the simulated DMSPd concentrations in the upper 10 m of the water column increased up to about 6 nmol L$^{-1}$ associated with the peak of simulated under-ice bloom (red line; Fig. 2b). This peak in simulated DMSPd was lower than the observed site-average value (ca. 11 nmol L$^{-1}$) from the second last sampling day. However, this observed site-average DMSPd value was associated with a large standard deviation because a single high value (ca. 30 nmol L$^{-1}$) was measured at 1.5 m

depth, while values measured deeper in the water column were much lower ($\leq 3$ nmol L$^{-1}$). Given this observed range, the simulated DMSPd peak is reasonable.

Figure 3c shows the simulated time series of the bottom-ice and the upper 10 m average of the water column DMS concentrations. The simulated bottom-ice DMS concentration increased gradually from April and reached about 1600 nmol L$^{-1}$ in late May (black line; Fig. 3c). The timing of the simulated DMS peak lagged behind the simulated peaks of DMSPp and DMSPd by about one week. While no DMS observations were available to directly compare with the simulated DMS for this time period, the simulated DMS peak is close to the DMS concentration of 2000 nmol L$^{-1}$ measured in the bottom ice in Resolute Passage at the end of the ice algal bloom in 2012 (Levasseur, 2013). The simulated bottom-ice DMS concentration remained close to its peak value until the beginning of June, then it quickly decreased to nearly 0 nmol L$^{-1}$ by late June.

In the upper 10 m of the water column, the simulated DMS concentrations increased gradually during early June and sharply during mid-June (red line; Fig. 3c). A few days after the simulated peaks of DMSPp and DMSPd in the upper water column, the simulated DMS reached its maximum value of about 9 nmol L$^{-1}$ in late June. This peak value is comparable to some of the high values of surface seawater DMS concentration measured in the eastern Canadian Archipelago and Baffin Bay during July and August of 2014 (Mungall et al., 2016).

### 3.1.4 Production and removal rates of DMSPd and DMS

The variability of the simulated DMSPd and DMS concentrations is driven by a range of physical and biogeochemical processes that are generally not well constrained by observations. Reporting the rates of those processes simulated by the model will help interpret the observed features. Figure 4a shows the individual terms in the production and removal rates of simulated bottom-ice DMSPd. Prior to mid-May, the simulated production rates by cell lysis and exudation increased to about 600 nmol L$^{-1}$ d$^{-1}$, associated with the simulated ice algal bloom. However, the two rates differed by twofold during the peak of simulated ice algal bloom. The production rate by cell lysis exceeded 1300 nmol L$^{-1}$ d$^{-1}$ as a result of increased ice algal biomass as well as nutrient stress in the bottom ice. On the other hand, the production rate by exudation remained around 600 nmol L$^{-1}$ d$^{-1}$ because its potential enhancement due to nutrient stress was offset by reduced primary production. The removal of simulated bottom-ice DMSPd was dominated by bacterial consumption, while the contributions of free DMSP-lyase and release from bottom ice were minor. As parameterized in the model, the removal rate by bacterial consumption varied with the bottom-ice DMSPd concentration, and peaked at 1800 nmol L$^{-1}$ d$^{-1}$ in late May. The simulated removal rates by free DMSP-lyase were generally low, reaching up to 50 nmol L$^{-1}$ d$^{-1}$) in late May. This value falls within the range observed in Antarctic sea ice brine samples (21-62 nmol L$^{-1}$ d$^{-1}$; Asher et al., 2011). The simulated removal rates by release from bottom ice reached up to about 200 nmol L$^{-1}$ d$^{-1}$ during the melt period. We note that the simulated removal rates by bacterial consumption and free DMSP-lyase both have the same functional form (Appendix A), and therefore the differences in these rates were a straightforward consequence of choices of parameter values.

Figure 4b shows the production and removal rates of simulated DMSPd in the uppermost layer (0.5 m below the ice) of the water column. In early June, the simulated production rates by release from bottom ice reached about 6 nmol L$^{-1}$ d$^{-1}$ and dominated the under-ice DMSPd budget as other terms were relatively small due to low biological activity under the ice.

During the simulated under-ice phytoplankton bloom in mid-June, both cell lysis and exudation made similar contributions (27-28 nmol L$^{-1}$ d$^{-1}$) to the DMSPd production in the uppermost layer of the water column, with the peak in cell lysis lagging a few days behind the peak in exudation. Finally, the simulated DMSPd production rates by sloppy feeding were negligible ($\leq 0.1$ nmol L$^{-1}$ d$^{-1}$) due to low zooplankton biomass during the melt period. No measurements of zooplankton biomass are available to observationally assess our simulated zooplankton biomass, although a previous study suggests high interannual variability in zooplankton biomass in Resolute Passage (Michel et al., 2006). The removal of simulated DMSPd in the uppermost layer of the water column was governed by bacterial consumption, which increased up to 35 nmol L$^{-1}$ d$^{-1}$ during the under-ice bloom. The simulated DMSPd removal rates by bacterial consumption were comparable to the rates measured under the ice in Resolute Passage during the melt period in 2012 (3 to 44 nmol L$^{-1}$ d$^{-1}$; Galindo et al., 2015). The simulated DMSPd removal rates by free DMSP-lyase were negligible (below 1 nmol L$^{-1}$ d$^{-1}$) in the uppermost layer of the water column throughout the simulated period.

Figure 4c shows the production and removal rates of simulated bottom-ice DMS. The production of simulated bottom-ice DMS was dominated by bacterial DMSPd-to-DMS conversion, while the production by free DMSP-lyase was considerably less. As parameterized in the model, the temporal variability of DMS production rates by bacterial conversion resembled the temporal variability of DMSPd removal rates by bacterial consumption. The simulated DMS production rates by bacterial conversion were highest (ca. 350 nmol L$^{-1}$ d$^{-1}$) during the peak of the ice algal bloom. The simulated DMS production rates by free DMSP-lyase increased gradually with the accumulation of DMSPd in the bottom ice (black line; Fig. 3b), but remained below 50 nmol L$^{-1}$ d$^{-1}$ throughout the simulated period. The removal of simulated DMS in the bottom ice was dominated by bacterial consumption, while photolysis and release from bottom ice became of comparable importance during the melt period. The simulated DMS removal rate by bacterial consumption reached about 325 nmol L$^{-1}$ d$^{-1}$ during the peak of ice algal bloom, balancing the DMS production by bacterial conversion. During the melt period, the simulated DMS removal rates by bacterial consumption were reduced due to a decrease in the bottom DMSPd (black line; Fig. 3b), while the removal rates by photolysis and release from bottom ice temporarily exceeded 50 nmol L$^{-1}$ d$^{-1}$. The increase in the simulated DMS removal rates by photolysis at the beginning of the melt period was caused by the increased light penetration through the ice. Despite the continuous melting of simulated snow and ice and the enhancement in light penetration, the removal rate by photolysis decreased sharply after its peak in early June due to the decrease in the bottom-ice DMS concentrations (black line; Fig. 3c). In mid-June, the simulated DMS removal rate by release from bottom ice reached its peak. This peak value was quantitatively comparable to the rates by other simulated processes at that time. Asher et al. (2011) measured gross DMS consumption rates in brine samples, which includes both rates of bacterial consumption and photolysis. Their reported values (57-250 nmol L$^{-1}$ d$^{-1}$) are generally comparable to our simulated values, although the peak values are beyond their reported range.

Figure 4d shows the production and removal rates of simulated DMS in the uppermost layer of the water column. Similarly to the simulated under-ice DMSPd budget, release from bottom ice dominated (>5 nmol L$^{-1}$ d$^{-1}$) the under-ice DMS budget prior to the under-ice bloom (i.e. early June). During the same time period, the simulated DMS production rates by bacterial conversion were relatively low (0-1 nmol L$^{-1}$ d$^{-1}$), which is consistent with the rates of 0-1.1 nmol L$^{-1}$ d$^{-1}$ measured in Resolute Passage during the initiation of the under-ice bloom in 2012 (Galindo et al., 2015). With the development of the

under-ice boom, the simulated DMS production rates by bacterial conversion increased quickly and reached a peak of ca. 7 nmol $L^{-1}$ $d^{-1}$ in mid-June. The simulated DMS production by free DMSP-lyase had a negligible contribution (<0.2 nmol $L^{-1}$ $d^{-1}$) throughout the simulation period, which is consistent with the rate measurement of ice-covered seawater samples in the Antarctic region (Asher et al., 2011). The removal process of DMS in the uppermost layer of the water column was

dominated by bacterial consumption which increased up to ca. 6 nmol $L^{-1}$ $d^{-1}$ during the under-ice bloom. The simulated DMS removal rates by photolysis also increased during the same time period, but they were relatively low (ca. 1 nmol $L^{-1}$ $d^{-1}$). The combined removal rates by bacterial consumption and photolysis are comparable to the rate measured in Antarctic ice-covered seawater (Fig. 3 of Asher et al., 2011). Finally, it is important to note that, in the standard run, the loss of DMS by sea-to-air flux was prevented due to the presence of ice under the assumption that the surface was fully ice-covered throughout

the simulation period. In Section 3.2.3, we will examine the effects of interstices in the ice on the simulated sea-to-air flux.

Due to the scarcity of rate measurements in ice-covered regions, it is undoubtedly challenging to evaluate the rates simulated by our model. The direct comparisons of those rates that have been measured in sea ice brines and under-ice seawater samples (i.e. DMS conversion, bacterial DMSPd/DMS consumption, free DMSP-lyase) indicate that our simulated rates are in good agreement with the observed rates (Asher et al., 2011; Galindo et al., 2015). Certainly, further rate measurements in ice-covered

regions will help build confidence in model-based estimates for production and removal rates of DMSPd and DMS within and under the sea ice. For simulated processes whose observed rates are not available, we find that the simulated rates in under-ice seawater (Figures 4b and d) are on the same order of magnitude as the observed rates in open-water environment (e.g. Galí and Simó, 2015), while the simulated rates in the bottom ice (Figures 4a and c) are a few orders of magnitude higher than those observed rates. Such results are expected for rates that are dependent on either DMSPd or DMS concentrations.

For example, bacterial consumption rates and photolysis rates are generally known to follow Michaelis-Menten kinetics (Galí and Simó, 2015, and references therein), whose rates can be represented as a product of some rate constant and the DMSPd (DMS) concentration. Since all of these rate constants prescribed in our model are based on field measurements in open water environment (see Appendix), the difference between the simulated rates in bottom-ice/under-ice environment and the observed rates in open-water environment reflects the difference in the concentration of DMSPd (DMS) in those environments which

differs by a few orders of magnitude.

## 3.2  Sensitivity runs

### 3.2.1  Incorporation of sea-ice biogeochemistry

In the standard run, we showed that the release of DMSPd and DMS from bottom ice respectively dominated the DMSPd and DMS budgets in the underlying water column prior to the onset of the under-ice phytoplankton bloom (Figs. 4b and 4d).

To evaluate the changes in the simulated under-ice DMSPd and DMS concentrations due to this release, we conducted an additional simulation that excluded the sea-ice sulfur cycle module (NoIceSul; Fig. S2b). In other words, DMSPd and DMS in the bottom ice are not simulated in the NoIceSul run, and therefore there is no release of these sulfur compounds from the bottom ice during the melt period. It should be emphasized, however, that the sea-ice ecosystem module was still retained in this

sensitivity run, hence the simulated ecosystem dynamics remained unchanged with respect to the standard run. The difference between the results of the standard and NoIceSul runs (i.e. Standard - NoIceSul; Fig. 5 and Table 2) thus represents the effect of the sea-ice sulfur cycle. As expected, the exclusion of the sea-ice sulfur cycle module resulted in a decrease in the under-ice DMSPd and DMS concentrations during most of the melt period (Fig. 5). The differences in these concentrations between the two runs were most evident from 1 June to 25 June, with peak differences of 0.5 (DMSPd) and 2.4 nmol $L^{-1}$ (DMS) during the third week of June. Following 25 June, the concentration differences between the two runs became negligible, as the release from bottom ice became small toward the end of the melt period (Fig. 4). Over the simulation period, the incorporation of the sea-ice sulfur cycle resulted in 6 and 18 % increases in the respective under-ice DMSPd and DMS pools (Table 2). The increase in DMS was much greater than that of DMSPd because the rates of increase in the under-ice DMS due to the release from bottom ice were relatively high among the under-ice DMS budget components, while the rates of increase in under-ice DMSPd due to the release from bottom ice were smaller compared to the other under-ice DMSPd budget components considered in the model.

Besides the sea-ice sulfur cycle, it is possible that the incorporation of the sea-ice ecosystem itself can have an impact on the under-ice DMSPd and DMS concentrations (e.g. via changes in the nutrient availability in the surface ocean). To examine this possibility, we conducted an additional simulation that excluded the entire sea-ice biogeochemical module (NoIceBgc; Fig. S2c), in which no release or uptake from sea ice is simulated. It is clear from Fig. 5 that the under-ice DMSPd and DMS concentrations simulated in the NoIceBgc run are much different from the results of the NoIceSul run, which implies a substantial contribution from the sea-ice ecosystem. In particular, the under-ice DMSPd and DMS concentrations in the NoIceBgc run were higher than the NoIceSul run in late June onward. The higher peaks in the under-ice DMSPd and DMS concentrations in the NoIceBgc run were associated with an under-ice phytoplankton bloom that was greater in magnitude than the bloom in the standard or the NoIceSul run (Fig. 6a). In the standard run, the presence of ice algae resulted in reduced under-ice nitrate concentrations due to uptake by the sea-ice ecosystem (Fig. 6b). Phytoplankton were able to increase biomass more in the NoIceBgc run because of the higher availability of nutrients. Consequently, the production rates of DMSPd and DMS associated with this bloom were also higher, thereby yielding higher DMSPd and DMS concentrations under the ice in the NoIceBgc run during the under-ice bloom.

On the other hand, prior to the under-ice bloom, the under-ice DMSPd and DMS concentrations were lower in the NoIceBgc run than in the standard run due to the lack of ice algal release (Fig. 6c), which seeded the under-ice bloom (Fig. 6a). Our results therefore suggest that the incorporation of the sea-ice ecosystem promotes the under-ice DMSPd and DMS production by seeding the under-ice phytoplankton bloom, while it reduces the overall production by drawing down the available nutrients prior to the bloom. Over the simulation period, the incorporation of the sea-ice ecosystem resulted in a 16 % decrease in the under-ice DMS concentrations relative to the NoIceBgc run (Table 2). Note that although the shading effect of ice algae likely contributed to a delay in the under-ice bloom, an earlier onset of the bloom in the NoIceBgc run relative to the standard run (Fig. 6a) suggests that the shading was less effective in modifying the bloom dynamics than the seeding. Furthermore, the effect of brine convection on nutrient dynamics (e.g. Vancoppenolle et al., 2010) was not taken into account in our model, which could further differentiate the results between the standard and the NoIceBgc runs.

The results presented here suggest that the incorporation of sea-ice biogeochemistry (referring to both sea-ice sulfur cycle and ecosystem) has both direct and indirect effects on the under-ice DMSPd and DMS production. The direct effect is due to the incorporation of sea-ice sulfur cycle which increases the under-ice DMSPd and DMS concentrations through the release of these sulfur species from the bottom ice. The indirect effect is due to the incorporation of sea-ice ecosystem which, depending on the phase of the under-ice phytoplankton bloom, increases or decreases the under-ice DMSPd and DMS concentrations by affecting the dynamics of the bloom. Over the simulation period, the incorporation of sea-ice biogeochemistry resulted in a slight change (-1 %) in the under-ice DMS concentrations, as the direct and indirect effects nearly counteracted each other (Table 2). However, the transient increases prior to the under-ice bloom peak (up to 5.6 nmol $L^{-1}$) could still be a significant source of episodic sea-air flux of DMS. We will examine the effects of these increases in the under-ice DMS production on the flux in Section 3.2.3.

### 3.2.2 Parameter uncertainty

The results of the standard run are influenced by the choice of uncertain model parameters. The model parameters of the sea-ice sulfur cycle are especially poorly constrained due to the scarcity of rate measurements within sea ice (Stefels et al., 2012). Therefore, it is important to report the sensitivity of our model results to plausible changes in these parameters. We conducted five additional simulations to examine the respective changes in the simulated DMS concentrations in the bottom ice and underlying water column due to a doubling of the following five key parameters: intracellular DMSP:Chl $a$ ratio (Case 1), DMS yield (Case 2), and rate constants for bacterial DMSPd consumption (Case 3), bacterial DMS consumption (Case 4), and photolysis (Case 5). Note that these parameter changes were applied only to the sea-ice sulfur cycle and not to the ocean sulfur cycle. We selected these five parameters considering that previous model sensitivity studies indicated their importance to marine sulfur cycle dynamics (Archer et al., 2004; Steiner and Denman, 2008).

The intracellular DMSP:Chl $a$ ratio is defined here as the ratio of particulate DMSP (DMSPp) to Chl $a$. For sea-ice samples, there are only three studies that report values of this ratio (Levasseur et al., 1994; Bouillon et al., 2002; Galindo et al., 2014), while several other studies provide the ratio of total DMSP (DMSPt = DMSPp + DMSPd) to Chl $a$ (Table 3). Our baseline value of 9.5 nmol:$\mu$g is taken directly from the Arctic-ICE study conducted in 2010. This value does not differ much from that obtained in the following year (9.4 nmol:$\mu$g; Galindo et al., 2014). In contrast, an earlier study in the same region gives a much lower value (2.7 nmol:$\mu$g; Levasseur et al., 1994), which is close to the mean ratio for pelagic diatoms (4 nmol:$\mu$g; Stefels et al., 2007). The reported DMSPt:Chl $a$ ratios for ice diatoms range from 8.4 nmol:$\mu$g to 49 nmol:$\mu$g (Table 3), which suggests that the DMSPp:Chl $a$ ratios could vary by a similar range. Note that most of these reported values are potentially underestimated due to anticipated DMSP loss associated with cell rupture during the melting process used in making these measurements (Stefels et al., 2012). Given this wide range among various studies and potential bias in measurements due to methodological challenges mentioned above, the doubled ratio of 19 nmol:$\mu$g in Case 1 was deemed reasonable in the natural environment. In Case 2, the DMS yield fraction was increased to 40 %, which was the upper limit of the measured range for the bottom section of Antarctic ice core samples (Stefels et al., 2012). The doubled DMS consumption rate constant of 0.4 $d^{-1}$ in Case 4 is within the range of 0.1-0.5 $d^{-1}$ observed in the bottom ice of Antarctic sea ice (J. Stefels, University of Groningen,

personal communication). To the best of our knowledge, there has been only one study measuring the bottom-ice parameter values tested in Cases 2 and 4, and no studies have measured the rate constants in the bottom ice considered in Cases 3 and 5. Doubling the values of these parameters is justified by the fact that the observed water column values of these parameters in the Arctic often differ by an order of magnitude (e.g., Luce et al., 2011; Galindo et al., 2015).

Figure 7 shows the simulated time series of bottom-ice and under-ice DMS concentrations in the standard and sensitivity runs. The results generally indicate that the parameter variations affected the magnitudes of the simulated DMS pools. The temporal patterns of DMS concentrations are more or less invariant, as they are controlled by the ecosystem dynamics. The parameter variations generally had greater impacts on the bottom-ice DMS concentrations than on those in the underlying water column (Table 4). For example, doubling the intracellular DMSP:Chl $a$ ratio (Case 1) and the DMS yield fraction (Case

2) resulted in doubling (100 % increase) and near-doubling (91 % increase) of the bottom-ice DMS, while the increases were lower (respectively 17 % and 12 %) in the uppermost layer of the water column. Nevertheless, doubling these parameters resulted in the largest change in the cumulative under-ice DMS among the five sensitivity runs. A previous model study by Lefèvre et al. (2002) also found these two parameters to be the most influential, and several other studies support the strong influence of variations in the intracellular DMSP:Chl $a$ ratio (Gabric et al., 1993; Archer et al., 2004; Steiner and Denman,

2008). Doubling the remaining parameters (Cases 3-5) had relatively small effects (<10 %) on the cumulative under-ice DMS. This result indicates that field measurements targeting the two most sensitive parameters (i.e. DMSPp:Chl $a$ ratio and DMS yield) will have the largest influence on constraining model-based estimates of sea-ice sulfur cycle processes.

    To a certain extent, these sensitivities (e.g. sign and/or relative magnitude of the change in DMS) could have been predicted from inspection of the model equations, in which it is evident that doubling a single parameter results in doubling the DMS

production or removal rate of a certain process either directly (Cases 2, 4, and 5) or indirectly (by doubling the production rate of precursor DMSPp; Case 1). The only exception is Case 3, in which doubling the bacterial DMSPd consumption rate constant affects both the production of DMS by bacterial conversion and the removal of DMSPd by bacterial consumption. Because the two processes have the opposite effect on the rate of change in DMS, it is even challenging to predict the sign (increase or decrease) of the change in DMS. The result of the sensitivity run indicates that the impact of doubling this parameter is a slight

decrease (2 %) both in the bottom- and under-ice DMS pools, and therefore the increases in the rates of the two processes are almost balanced. Although we cannot explain the magnitude of the percentage change in the DMS pools, we are confident that the net effect of doubling the bacterial DMSPd consumption rate constant on the DMS pools is a decrease because the model considers that only a fraction (i.e. the DMS yield fraction; set to 0.2 in this run) of DMSPd consumed by bacteria is converted to DMS, while the remaining fraction is lost to the sulfur pool. It is of particular importance to investigate the

model sensitivity to variations in a parameter that has influence on multiple processes of the sulfur cycle, such as the bacterial DMSPd consumption rate constant. Although this is beyond the scope of the present study, it is worthwhile to mention that two parameters of ecological processes, namely active exudation fraction and cell lysis rate constant, have influence on both ecological and sulfur processes, and therefore deserve attention in future sensitivity studies.

### 3.2.3 Sea-air DMS flux during the melt period

In the uppermost layer of the water column, the sea-air fluxes of DMS ($\mu$mol m$^{-2}$ s$^{-1}$) were calculated as a function of areal fraction of open water ($f_{ow}$), gas transfer velocity ($k_{dms}$), and the concentration of seawater DMS ($DMS_{wc}$):

$$Flux = f_{ow} k_{dms} DMS_{wc} \tag{1}$$

This formulation assumes that the atmospheric DMS concentration is sufficiently lower than the seawater value, such that it can be neglected for computing the gradient. This assumption is common in both measurement- and model-based estimates of oceanic DMS flux (e.g. Rempillo et al., 2011; Tesdal et al., 2016). The gas transfer velocity is parameterized following Nightingale et al. (2000). Note that this parameterization is based on measurements in open waters, therefore may not be suitable for ice-covered waters. However, we used the Nightingale et al. (2000) parameterization in order to better compare

with previous flux estimates in the ice-covered Arctic that are all based on similar parameterizations. Future studies should take into account the effects of ice-associated processes on gas transfer velocity parameterizations (e.g. Loose et al., 2014). Also note that we do not take into account additional fluxes from other surface types, such as snow, bare ice, and melt ponds, which may provide an additional source for atmospheric DMS (Zemmelink et al., 2008; Nomura et al., 2012; Levasseur, 2013; Mungall et al., 2016).

In the standard run, it was assumed that when sea ice was present, the surface was fully ice-covered and $f_{ow}$ was set to zero. Although this assumption is reasonable when conducting simulations at a single point in space, it is less reasonable over an entire grid cell due to heterogeneity in the subgrid-scale structure of surface fields. In fact, as suggested by Levasseur et al. (1994), sea-air DMS fluxes can take place through openings in the ice (such as leads and cracks) and at the ice margin. Furthermore, laboratory, field, and model studies have suggested that fluxes of $CO_2$ through small scale areas of open water

result in non-negligible fluxes in ice-covered regions (Loose et al., 2011; Else et al., 2012; Steiner et al., 2013). In order to quantify the potential emissions of DMS through the open water in an ice-covered area, we conducted four additional standard runs with non-zero $f_{ow}$ values (Table 5). In the first and second runs, the values of 0.02 and 0.1 were selected to represent small and large leads within the ice (Lindsay and Rothrock, 1995; Steiner et al., 2013). In the third run, the value of 0.5 was prescribed to represent either an extensive opening in the ice or emissions near the ice margin (such that only a half of under-ice

DMS can be advected to the ice margin and make its way into the atmosphere). Finally in the fourth run, the value of 1 was assigned to represent emissions right at the ice margin. Note that while these sensitivity runs are highly idealized (assuming partial or no ice cover for estimates of sea-air flux, but full ice cover in the biogeochemical model), they provide an indication on the impacts of open-water fractions on the temporal variability of the DMS flux. Also, $f_{ow}$ is included only in the DMS-flux parameterization, and has no influence on other physical or biogeochemical processes (such as surface heat fluxes). In order to

evaluate the contribution of sea-ice biogeochemistry to the simulated flux, these four runs with non-zero $f_{ow}$ values were also conducted for the NoIceSul and NoIceBgc cases.

During the melt period, observed winds were generally low to moderate, ranging on a daily average from 1 to <10 m s$^{-1}$ (Fig. 8). However, occasional strong winds were also measured as indicated by daily maximum wind speeds exceeding 20 m s$^{-1}$. The time series of sea-air DMS fluxes simulated by the standard run using four different values of $f_{ow}$ (Fig. 6b) were

generally high in late June, some of which coincided with these days of stronger winds as well as with peaks in under-ice DMS (Fig. 6b). In particular, the simulated fluxes were notably high on 16, 21, and 26 June, producing three distinct peaks in the time series. In the cases of emissions through partially open-water ($f_{ow}$ = 0.02, 0.1, and 0.5), the simulated maximum fluxes (of up to 0.3, 1.2, and 4.9 $\mu$mol m$^{-2}$ d$^{-1}$, respectively; Table 5) were higher than the observational upper-end flux estimates over regions of similar open-water fractions during July and August of 1994 (0.1 $\mu$mol m$^{-2}$ d$^{-1}$ for $f_{ow}$ = 0.03-0.06 and 1.2 $\mu$mol m$^{-2}$ d$^{-1}$ for $f_{ow}$ = 0.25-0.3; Sharma et al., 1999), most probably because these simulated maxima resulted partly from the peak in DMS associated with the under-ice bloom. In the cases of emissions near and at ice margins ($f_{ow}$ = 0.5 and 1), the simulated maxima (of up to 4.9 and 8.1 $\mu$mol m$^{-2}$ d$^{-1}$, respectively; Table 5) were comparable to the emissions under ice-free conditions estimated from previous oceanographic cruises in the Arctic (Leck and Persson, 1996; Sharma et al., 1999; Mungall et al., 2016). Furthermore, these simulated maxima exceeded the nucleation threshold of 2.5 $\mu$mol m$^{-2}$ d$^{-1}$, above which the DMS flux has been suggested to be sufficiently high to promote new particle formation in pristine marine conditions (Pandis et al., 1994; Russell et al., 1994).

As expected, simulated sea-air fluxes were smaller in the NoIceSul run than in the standard run for each of the four $f_{ow}$ values (Fig. 9b). The incorporation of sea-ice sulfur cycle affected the simulated fluxes most prominently during the first three weeks of June (Fig. 9d). The increase in the fluxes in the standard run relative to the NoIceSul run during this time period was due to the increase in the under-ice DMS through the release of bottom-ice DMS as discussed in Section 3.2.1. The relative flux enhancement was particularly important during the first two weeks of June, during which the simulated fluxes would otherwise remain close to zero as shown in Fig. 9b. During the third week of June, the first and second simulated spikes in the flux time series were increased by as much as 1.7 $\mu$mol m$^{-2}$ s$^{-1}$ in the case of $f_{ow}$ = 1 (Table 5). Overall, the incorporation of sea-ice sulfur cycle resulted in a 20-26 % DMS flux enhancement.

When both the sea-ice sulfur cycle and ecosystem modules were excluded from the model (NoIceBgc), the simulated flux time series fluctuated quite differently from those of the NoIceSul runs (Fig. 9c), implying an active contribution from sea-ice ecosystem to the simulated flux. The flux difference between the two runs indicates that the incorporation of sea-ice ecosystem results in an enhancement of fluxes between 13 and 19 June, followed by a reduction from 19 June onward (Fig. 9e). The three simulated spikes in the flux time series of both the standard and NoIceSul runs were all affected by the incorporation of sea-ice ecosystem: the first spike was enhanced by as much as 3.5 $\mu$mol m$^{-2}$ s$^{-1}$ (in the case of $f_{ow}$ = 1), while the second and third spikes were reduced by the similar amount. These changes in fluxes were primarily driven by the changes in under-ice DMS concentrations (Fig. 5b). Overall, the incorporation of sea-ice ecosystem resulted in a 9-14 % reduction in the simulated flux relative to the NoIceBgc run (Table 5).

Lastly, the overall effects of incorporating sea-ice biogeochemistry on simulated DMS fluxes were examined by calculating the flux difference between the standard and NoIceBgc runs (Fig. 9f). The largest positive flux difference occurring in the third week of June was due to the incorporation of both sea-ice sulfur cycle and ecosystem (Figs. 9d and e). This flux difference resulted in an enhancement of the first simulated spike by as much as 5 $\mu$mol m$^{-2}$ s$^{-1}$ (in the case of $f_{ow}$ = 1). On the other hand, the largest negative flux difference coincided with the occurrence of the third simulated spike, resulting in a reduction in this spike by nearly 4 $\mu$mol m$^{-2}$ s$^{-1}$ (in the case of $f_{ow}$ = 1). Over the simulation period, the incorporation of the sea-

ice biogeochemistry resulted in a 3-15 % flux enhancement (Table 5). Considering that the overall change in the under-ice DMS due to the incorporation of sea-ice biogeochemistry was only -1 % (Section 3.2.1), this result demonstrates the potential importance of episodic fluxes (such as those spikes simulated in this study) to the cumulative DMS flux.

## 3.3   Limitations of the present study

The model used in the present study incorporated many of the important physical and biogeochemical processes within and under the sea ice (Fig. 1). However, there are additional processes that may potentially be important to the sea-ice sulfur cycle but are neglected in the present study. These processes are shown schematically in Fig. 10. We will discuss some of the challenges in implementing these processes in order to help guide further advances in sea-ice sulfur cycle studies.

First, the effects of brine drainage on the release of biogeochemical state variables in the bottom ice were neglected in the
model, which possibly led to mismatches between our simulated and observed phytoplankton biomass, DMSPp, and DMSPd under the ice prior to the snowmelt period in early May (Section 3.1). Existing parameterizations for brine drainage typically require the model representation of brine dynamics (e.g. Vancoppenolle et al., 2007, 2010). While a simplified brine drainage term was included in our model salinity calculation (following Vancoppenolle et al., 2009), implementing such parameterizations into our biogeochemical model was not desirable, as the goal of the present modelling excercise was to develop parame-
terizations that can be easily implemented into 3-D sea ice models, many of which do not explicitly simulate brine dynamics. In addition, the implementation of brine drainage effects on biogeochemical tracers will require modifications compared to the salinity formulations. Although our test run showed that adding these effects resulted in an improvement in the temporal patterns of ice algal biomass and DMSP (not shown), we excluded these parameterizations from the current presentation as the parameters are at this point poorly constrained (e.g. vertically averaged sea ice equilibrium tracer concentrations during brine
drainage; see Eq. 19 of Vancoppenolle et al., 2009).

The production of DMSPd by sloopy feeding was neglected in the present study, as zooplankton grazing on ice algae was not considered in the ecosystem model (Mortenson et al., 2016). Although previous field measurements provide evidence for zooplankton grazing on ice algae (Michel et al., 1996, and references therein), the strength of this process during the Arctic-ICE 2010 study is unknown due to the absence of zooplankton measurements. The previous sea-ice sulfur cycle model study
by Elliott et al. (2012) suggests sloppy feeding as an important process for the DMSPd (and the subsequent DMS) production during the early stages of ice algal blooms. This argument is supported by a recent Antarctic ice study that found the DMSP content in krill specimens (Damm et al., 2016). However, note that the DMSPd production by sloppy feeding in Elliott et al. (2012) and that of exudation in our study are parameterized similarly (both parameterizations have a linear dependence on simulated ice algal growth rate) and DMSPd production by exudation was neglected in Elliott et al. (2012). This indicates
that model parameterizations might account for different processes in a similar way, accounting for a required source without observational evidence on details of the processes. Future observational studies are required to assess the relative importance of these two processes as well as the details on how they should be parameterized.

The production of DMS by dimethylsulfoxide (DMSO) reduction is not considered due to the lack of observational constraints on this process. This process has been suggested as a major pathway of DMS production in Antarctic sea ice (Asher

et al., 2011). The fact that Asher et al. (2011) is the only study that has measured the rate constants of DMSO reduction in sea-ice brines indicates the need for further observational studies.

A possible direct release of DMS by intracellular or extracellular DMSP-lyase activity of algae (Niki et al., 2000; Stefels, 2000; Alcolombri et al., 2015) is also disregarded. To the best of our knowledge, no studies as of yet have shown that diatoms, the dominant group of the bottom-ice algal community, possess or use DMSP-lyases. Thus, neglecting these processes seems plausible at least for bottom-ice sulfur cycle studies.

Recent studies on Arctic sea ice have shown that the gas bubble formation and rise plays a dominant role in the dynamics of an inert gas (argon) within sea-ice brines when sea ice becomes permeable (Zhou et al., 2013; Moreau et al., 2014). Given the fact that DMS can be present in gaseous phase, this process may be an relevant sink for DMS present within brine channels. The present study neglected this process as no observations are available to constrain a respective parameterization for DMS. Nonetheless, a complementary parameterization based on known differences in gas dynamics could be tested on the model as a future project.

Besides the sulfur cycle module, our model can be improved by further development of parameterizations for physical and ecological processes. For example, nutrient dynamics near the ice-water interface can be represented more realistically by explicitly resolving the role of brine convection (Vancoppenolle et al., 2010). Explicit representation of zooplankton grazing on ice algae would further advance the sea-ice ecosystem module. The refreezing of snow and the subsequent formation of superimposed ice were not simulated, although they were observed toward the end of the ArcticICE 2010 campaign (not shown). These processes have impacts on sea ice thermodynamics and light transfer through sea ice that indirectly affect the sea-ice sulfur cycle dynamics.

Lastly, we acknowledge the limitations of the assessment of model simulations in the present study. The model results were evaluated against observations coming from a single time-series study, and therefore, lacking the assessment of model's portability (i.e. the model's ability to simulate simulate the observed time series from a different year and/or region). The scarcity of field measurements of dimethylated sulfur compounds within sea ice makes this task extremely challenging. Clearly, more field measurements, and particularly, high-resolution time-series of dimethylated sulfur compounds within sea ice (e.g. Tison et al., 2010; Galindo et al., 2014, 2015; Carnat et al., 2016) are needed to assess the portability of 1-D sea-ice sulfur cycle models and further constrain the model parameters and parameterizations.

## 4   Conclusions

In the present study, we investigated the implications of sea-ice biogeochemistry for the oceanic production and emissions of DMS in the Arctic. Our model is able to capture reasonably well the limited set of observational data available, and suggests that sea-ice sulfur cycle and ecosystem have considerable impacts on the DMS production under the ice, and therefore should not be overlooked in the estimates of oceanic DMS fluxes especially near the ice margins. Specifically, the sea-ice sulfur cycle enhanced the under-ice DMS production directly by the release of bottom-ice DMSPd and DMS into the underlying water column, while the production was enhanced as well as reduced by interactions with the sea-ice ecosystem at various phases of

the under-ice phytoplankton bloom. In the case of first-year landfast ice in Resolute Passage, we estimated that the incorporation of sea-ice sulfur cycle resulted in a 18 % enhancement of DMS concentrations under the ice and a 20-26 % enhancement of sea-air DMS fluxes during the melt period. In contrast, the incorporation of a sea-ice ecosystem resulted in an overall reduction in the under-ice DMS production (16 %) as well as its emissions towards the atmosphere (9-14 %). The overall effect of sea-ice biogeochemistry (i.e. both sulfur cycle and ecosystem) appears to be nearly nil for the under-ice DMS production (-1 %), while it is an enhancement for the emissions (8-20 %). Furthermore, in the vicinity of ice margins, the simulated spikes in sea-air fluxes of DMS originating from the bottom ice and underlying water column were comparable to some of the local maxima in the summertime flux estimated for ice-free waters in the Arctic. We acknowledge the simplified representation of complex reality of the sea-ice sulfur cycle dynamics considered in our model, and note that the results of model simulations are subject to uncertainty owing to uncertainties in the model parameters and parameterizations. Furthermore, our model results are representative for a particular year, location, and specific environmental conditions. A few suggestions for future model development are to: 1) incorporate the state dependence of bacterial parameters that are deemed important to the sea-ice sulfur cycle dynamics, such as the bacterial DMS yield fraction; and 2) parameterize the effects of relevant processes, such as brine drainage, gas bubble release, sloppy feeding, and DMSO reduction.

To improve model-based estimates of oceanic DMS emissions in the Arctic under present-day and future climates, we make the following recommendations for future studies: both the sea-ice sulfur cycle and ecosystem should be incorporated into model simulations at a regional (pan-Arctic) scale; and more field time-series measurements of dimethylsulated sulfur compounds and key parameters of sea-ice sulfur cycle within sea ice (i.e. DMSPp:Chl $a$ ratio and DMS yield fraction) should be conducted to further assess the model performance and refine the representation of essential processes. Ultimately, ocean circulation models incorporating both the sea-ice sulfur cycle and ecosystem should be coupled to atmospheric chemistry transport models in order to explore the possibility of regional climate regulation by oceanic DMS emissions within the Arctic.

## Appendix A: Detailed model description

The set of differential equations describing the temporal evolution of DMSPp, DMSPd, and DMS concentrations in the bottom ice and water column is provided below. The list of variables and parameters involved in the model is provided in Table 1.

## A1 Sea-ice sulfur cycle

The meltwater equivalent concentration (nmol L$^{-1}$) of the particulate phase of DMSP in the bottom ice ($DMSPp$) is simulated diagnostically by assuming a fixed DMSPp-to-chlorophyll $a$ intracellular ratio (nmol S:$\mu$g Chl $a$) for ice algae ($q$):

$$\frac{\partial}{\partial t}(DMSPp) = q\frac{\partial}{\partial t}(IA) \tag{A1}$$

where $IA$ is the ice algal biomass ($\mu$g Chl $a$ L$^{-1}$). The intracellular ratio varies among algal species (Keller, 1989; Matrai and Keller, 1994) and also varies with various abiotic factors including temperature (Karsten et al., 1992; van Rijssel and Gieskes, 2002), salinity (Karsten et al., 1992), light (Karsten et al., 1992; Stefels and van Leeuwe, 1998; Sunda et al., 2002; Archer

et al., 2009; Galindo et al., 2016) and nutrients (Stefels and van Leeuwe, 1998; Sunda et al., 2007; Archer et al., 2009). The reported values for $q$ vary from 7.7 (Kirst et al., 1991) to about 20 (Uzuka, 2003; Stefels et al., 2012). Values of 9.4-9.5 have been reported for Resolute Passage (Galindo et al., 2014). In the standard run, we set $q$ to 9.5 nmol S:$\mu$g Chl $a$.

The meltwater equivalent concentration (nmol L$^{-1}$) of the dissolved phase of DMSP in the bottom ice ($DMSPd$) is simulated prognostically:

$$\frac{\partial}{\partial t}(DMSPd) = F_{lysis} + F_{exudation} - F_{consumption}^{dmspd} - F_{free} - F_{release}^{dmspd} \tag{A2}$$

where $F$ denotes the production or removal rate (nmol L$^{-1}$ d$^{-1}$) for each of the processes considered in the model (Fig. 1). The first two terms in Eq. (A2) represent the production rates of bottom-ice DMSPd by cell lysis and exudation, respectively. Following Archer et al. (2004), these processes are parameterized to increase under nutrient stress:

$$F_{lysis} = \frac{1}{L_{nut} + 0.1} k_{lysis} DMSPp \tag{A3}$$

$$F_{exudation} = [f_{active} + (1 - f_{active})(1 - L_{nut})] \mu DMSPp \tag{A4}$$

where $L_{nut}$ represents the nutrient limitation index (-) for ice algal growth, $k_{lysis}$ represents the rate constant (d$^{-1}$) for cell lysis, $f_{active}$ represents the active exudation fraction (-), and $\mu$ represents the ice algal specific growth rate (d$^{-1}$). Both $L_{nut}$ and $\mu$ are calculated by the ecosystem module. The two parameters involving cell lysis ($k_{lysis}$) and exudation ($f_{active}$) are generally poorly constrained in the sulfur cycle models because the measurements of production rates of DMSPd by cell lysis and exudation are very limited in seawater (Laroche et al., 1999). To our best knowledge, these rates have not been measured within sea ice. In the standard run, $k_{lysis}$ and $f_{active}$ are respectively set to 0.03 d$^{-1}$ and 0.05, which are similar to the values used in previous ocean sulfur cycle models (Archer et al., 2004; Steiner and Denman, 2008).

The third term in Eq. (A2) represents the removal rate of bottom-ice DMSPd by bacterial consumption. DMSPd is an important source of carbon and sulfur for bacteria in the marine environment, as the bacterial consumption of DMSPd can account for up to 15 % of their total carbon demand and almost all of their sulfur demand (Stefels et al., 2007). In the model, this removal process is parameterized as:

$$F_{consumption}^{dmspd} = k_{dmspd} DMSPd \tag{A5}$$

where $k_{dmspd}$ represents the rate constant (d$^{-1}$) for bacterial consumption of DMSPd. There are no reported values for $k_{dmspd}$ in sea ice. In the standard run, $k_{dmspd}$ is set to 1 d$^{-1}$ based on the model calibration.

The fourth term in Eq. (A2) ($F_{free}$) represents the removal rate of DMSPd by free DMSP-lyase present in the bottom ice. This process is parameterized as the product of a rate constant ($k_{free}$ [d$^{-1}$]) and the concentration of bottom-ice DMSPd:

$$F_{free} = k_{free} DMSPd \tag{A6}$$

In previous model studies, this process was considered as a minor removal pathway of DMSPd with $k_{free}$ varying from 0.01 d$^{-1}$ (Archer et al., 2004) to 0.04 d$^{-1}$ (Steiner et al., 2006). In the standard run, $k_{free}$ is set to 0.02 d$^{-1}$.

The fifth term in Eq. (A2) ($F_{release}^{dmspd}$) represents the removal rate of DMSPd due to its release into the underlying water column. Various processes, including gravity drainage, flushing, brine expulsion, flooding, and basal melting, can account for vertical movement of tracers within the sea ice brine channel. Parameterizations of these processes are very complex (e.g. Vancoppenolle et al., 2007), although simpler approaches have also been taken to represent these processes in previous sea

ice biogeochemical modelling studies (e.g., Tedesco and Vichi, 2014; Watanabe et al., 2015). By adopting a simpler approach similar to those previous studies, we parameterized the release resulting from two processes: 1) flushing due to drainage of snow meltwater accumulated in melt ponds and meltwater of surface and interior ice; and 2) sloughing due to basal melting. Specifically, the transfer velocity of flushing due to melt pond drainage is proportional to the area fraction ($A_{mp}$) and the drainage rate ($r_{mp}$) of melt ponds, while that of flushing of surface and interior ice meltwater and sloughing of basal ice is

proportional to the rate of change (decrease) in sea ice thickness ($\min(0, \frac{dh_i}{dt})$). The removal rate due to the release is then calculated by multiplying the total (i.e. flushing and sloughing) transfer velocity by the concentration of bottom-ice DMSPd:

$$F_{release}^{dmspd} = \frac{1}{h_{bi}} \min\left(0, \frac{\rho_i}{\rho_{me}} \frac{dh_i}{dt} - A_{mp}r_{mp}\right) DMSPd \tag{A7}$$

where $h_{bi}$ represents the thickness of the bottom ice skeletal layer, which is set to 0.03 m. The ratio of sea ice to meltwater densities accounts for the volume difference between sea ice and meltwater, which are set respectively to 913 and 1000 kg

m$^{-3}$. $A_{mp}$ and $\frac{dh_i}{dt}$ are computed by the physical model (Flato and Brown, 1996; Abraham et al., 2015). A constant drainage rate of 0.0175 m d$^{-1}$ is prescribed to $r_{mp}$ following (Taylor and Feltham, 2004).

The meltwater equivalent concentration (nmol L$^{-1}$) of DMS in the bottom ice ($DMS$) is simulated prognostically:

$$\frac{\partial}{\partial t}(DMS) = F_{conversion} + F_{free}^{dms} - F_{consumption}^{dms} - F_{photolysis} - F_{release}^{dms} \tag{A8}$$

The first term in Eq. (A8) ($F_{conversion}$) represents the production rate of bottom-ice DMS by bacterial conversion of DMSPd

to DMS. This process is one of the two major degradation pathways for DMSPd consumed by bacteria in open waters. The bacteria cleave DMSPd and yield DMS along with other products such as acrylate and a proton (Stefels et al., 2007). The other major degradation pathway is known as demethylation/demethiolation (Kiene and Linn, 2000), which is accounted for in the model as part of the DMSPd removal rate by bacterial consumption. The rate of DMS production via bacterial DMSPd conversion is often scaled to the bacterial consumption rate of DMSPd, such that the former can be expressed as a fraction of

the latter:

$$F_{conversion} = f_{yield}F_{consumption}^{dmspd} \tag{A9}$$

where $f_{yield}$ is known as the DMS yield fraction (-). Only one study has reported values for $f_{yield}$ measured in the bottom ice, all less than 0.4 (Stefels et al., 2012). In the standard run, $f_{yield}$ is set to 0.2.

The second term in Eq. (A8) ($F_{free}$) represents the production rate of bottom-ice DMS via free DMSP-lyase which is

equivalent to the fourth term in Eq. (A2) and is defined in Eq. (A6).

The third term in Eq. (A8) ($F_{consumption}^{dms}$) represents the removal rate of bottom-ice DMS by bacterial consumption which is parameterized similarly to the bacterial consumption of bottom-ice DMSPd (Eq. (A5)):

$$F_{consumption}^{dms} = k_{dms} DMS \tag{A10}$$

where $k_{dms}$ represents the rate constant ($d^{-1}$) for bacterial consumption of DMS. The only measurements of $k_{dms}$ conducted for ice core samples showed a range from 0.1 to 0.5 $d^{-1}$ for the bottom ice (J. Stefels, University of Groningen, personal communication). In the standard run, $k_{dms}$ is set to 0.2 $d^{-1}$.

The fourth term in Eq. (A8) ($F_{photolysis}$) represents the removal rate of bottom-ice DMS by photolysis, a photochemical process that converts DMS into its oxidation product, DMSO. The rate of photolysis is primarily determined by ambient light conditions, particularly in the ultraviolet (UV) wavelengths (Toole et al., 2004). However, we do not incorporate the UV dependence on the photlysis parameterization as the model does not have a representation for UV. Instead, we parameterize the light dependence of $F_{photolysis}$ using the photosynthetically active radiation similarly to Archer et al. (2004):

$$F_{photolysis} = k_{photolysis} \frac{PAR}{PAR + h_{photolysis}} DMS \tag{A11}$$

where $PAR$ represents the photosynthetically active radiation reaching the bottom ice (W $m^{-2}$), which is computed by the physical model. The parameters, $k_{photolysis}$ and $h_{photolysis}$ represent the rate constant ($d^{-1}$) and the half-saturation constant (W $m^{-2}$) for photolysis in the bottom ice. To the best of our knowledge, no studies have reported the values for photolysis rate constant in the bottom ice. In the standard run, $k_{photolysis}$ is set to 0.1 $d^{-1}$ based on the measurements in the water column (discussed in Sec. A2). We assume that photolysis is inhibited under low light conditions, and therefore set $h_{photolysis}$ to 1 W $m^{-2}$.

The last term in Eq. (A8) ($F_{release}^{dms}$) represents the removal rate of bottom-ice DMS due to flushing and melting, which is parameterized in the same way as $F_{release}^{dmspd}$:

$$F_{release}^{dms} = \frac{1}{h_{bi}} \min\left(0, \frac{\rho_i}{\rho_{me}} \frac{dh_i}{dt} - A_{mp} r_{mp}\right) DMS \tag{A12}$$

## A2 Ocean sulfur cycle

The concentration (nmol $L^{-1}$) of particulate DMSP in the water column ($DMSPp_{wc}$) is simulated diagnostically by assuming a fixed DMSPp-to-chlorophyll $a$ intracellular ratio (nmol S:$\mu$g Chl $a$) for each phytoplankton group ($q_{p1}$ and $q_{p2}$):

$$\frac{\partial}{\partial t}(DMSPp_{wc}) = q_{p1} \frac{\partial}{\partial t} P1 + q_{p2} \frac{\partial}{\partial t} P2 \tag{A13}$$

where $P1$ and $P2$ represent the biomass of small and large phytoplankton ($\mu$g Chl $a$ $L^{-1}$), respectively. Although the model does not specify the species group for $P1$, it is assumed that $P1$ produces more DMSP for a given amount of chlorophyll $a$ than diatoms ($P2$). In the standard run, $q_{p1}$ is set to 100 which is close to the intracellular ratios for non-diatom species groups reported in Stefels et al. (2007), and $q_{p2}$ is set to 9.5 which is equivalent to the intracellular ratio for ice algae ($q$).

The concentration (nmol L$^{-1}$) of DMSPd in the water column ($DMSPd_{wc}$) is simulated prognostically:

$$\frac{\partial}{\partial t}(DMSPd_{wc}) = F_{lysis}^{wc} + F_{exudation}^{wc} + F_{sloppy}^{wc} + F_{icesea}^{dmspd.wc} - F_{consumption}^{dmspd.wc} - F_{free}^{wc}$$

$$+ \frac{\partial}{\partial z}\left(K_z \frac{\partial}{\partial z}(DMSPd_{wc})\right) \tag{A14}$$

where the last term represents the mixing rate of DMSPd between model layers, with $K_z$ being the vertical eddy diffusivity (m$^2$s$^{-1}$) which is calculated by the ocean physical model.

The first, second, fifth, and sixth terms in Eq. (A14) ($F_{lysis}^{wc}$, $F_{exudation}^{wc}$, $F_{consumption}^{dmspd.wc}$, $F_{free}^{wc}$) are parameterized similarly to those in the sea-ice sulfur cycle (Eq. (A2)):

$$F_{lysis}^{wc} = \frac{1}{L_{nut}^{p1}+0.1}k_{lysis}^{p1}q_{p1}P1 + \frac{1}{L_{nut}^{p2}+0.1}k_{lysis}^{p2}q_{p2}P2 \tag{A15}$$

$$F_{exudation}^{wc} = \left[f_{active}^{p1} + \left(1-f_{active}^{p1}\right)\left(1-L_{nut}^{p1}\right)\right]\mu_{p1}q_{p1}P1 + \left[f_{active}^{p2} + \left(1-f_{active}^{p2}\right)\left(1-L_{nut}^{p2}\right)\right]\mu_{p2}q_{p2}P2 \tag{A16}$$

$$F_{consumption}^{dmspd.wc} = k_{dmspd}^{wc}DMSPd_{wc} \tag{A17}$$

$$F_{free}^{wc} = k_{free}^{wc}DMSPd_{wc} \tag{A18}$$

where the nutrient limitation indices ($L_{nut}^{p1}$ and $L_{nut}^{p2}$) and the growth rates ($\mu_{p1}$ and $\mu_{p2}$) of small and large phytoplankton, respectively, are calculated by the ocean ecosystem module. To our best knowledge, there are no reported values for the cell lysis rate constants ($k_{lysis}^{p1}$ and $k_{lysis}^{p2}$) and the active exudation fractions ($f_{active}^{p1}$ and $f_{active}^{p2}$) in ice-covered regions. In the standard run, $k_{lysis}^{p1}$ and $k_{lysis}^{p2}$ are set to 0.03 d$^{-1}$, and $f_{active}^{p1}$ and $f_{active}^{p2}$ are set to 0.05 for both small and large phytoplankton. $k_{dmspd}^{wc}$ represents the bacterial DMSPd consumption rate constant in the water column. The reported values for $k_{dmspd}^{wc}$ in Arctic surface water vary from 1.5 d$^{-1}$ during autumn (Luce et al., 2011; Motard-Côté et al., 2012) to 4.1 d$^{-1}$ during spring (Galindo et al., 2015). In the standard run, $k_{dmspd}^{wc}$ is set to 5 d$^{-1}$) based on the model calibration. $k_{free}^{wc}$ represents the rate constant for free DMSP-lyase in the water column, which is set to 0.02 d$^{-1}$ in the standard run.

The third term in Eq. (A14) represents the production rate of DMSPd in the water column by sloppy feeding:

$$F_{sloppy} = f_{sloppy}^{z1}q_{p1}R_{p1}^{z1} + f_{sloppy}^{z2}q_{p2}R_{p2}^{z2} \tag{A19}$$

where $f_{sloppy}^{z1}$ and $f_{sloppy}^{z2}$ represent the fractions of sloppy feeding by small and large zooplankton. In the standard run, these fractions are set to 0.3 for both zooplankton groups, based on the findings that 20 to 70 % of grazed DMSPp is released into the ambient seawater as DMSPd (Stefels et al., 2007). $R_{p1}^{z1}$ and $R_{p2}^{z2}$ represent the loss rates ($\mu$g Chl $a$ L$^{-1}$ d$^{-1}$) of small and large phytoplankton due to grazing by small and large zooplankton, respectively, which are calculated by the ecosystem module.

The fourth term in Eq. (A14) represents the rate of change in under-ice DMSPd due to exchanges of DMSPd between the basal ice and underlying seawater and concentration (dilution) as a result of ice growth (melting), which can be written in the vertically discretized form as:

$$F_{icesea}^{dmspd.wc} = \frac{1}{h_{z_0}}\left(\frac{\rho_i}{\rho_{wc}}\frac{dH_i}{dt} - \frac{\rho_{me}}{\rho_{wc}}A_{mp}r_{mp}\right)\left(DMSPd_{wc} - \frac{\rho_{me}}{\rho_{wc}}DMSPd^*\right)\delta_{z,z_0} \tag{A20}$$

where $h_{z_0}$ is the thickness of the uppermost layer of the water column (set to 1 m) and $\rho_{wc}$ is the density of seawater (calculated by the physical model). Again, the ratios of densities account for the volume differences in order to calculate the rate in seawater equivalent. $DMSPd^*$ represents the meltwater equivalent concentration of DMSPd taken up by the ice during ice growth, or released into the water column during melting. We make the assumption that, unlike salt, DMSPd taken up during ice growth is sufficiently small, such that $DMSPd^*$ can be set to zero. During the flushing and melting periods, $DMSPd^*$ is set to DMSPd in the bottom ice. The Kronecker's delta ($\delta_{z,z_0}$) equals 1 at the uppermost layer of the water column ($z_0$), whereas it is 0 elsewhere.

The concentration (nmol L$^{-1}$) of DMS in the water column ($DMS_{wc}$) is simulated prognostically:

$$\frac{\partial}{\partial t}(DMS_{wc}) = F^{wc}_{conversion} + F^{wc}_{free} + F^{dms.wc}_{icesea} - F^{dms.wc}_{consumption} - F^{wc}_{photolysis} - F_{seaair} + \frac{\partial}{\partial z}\left(K_z \frac{\partial}{\partial z}(DMS_{wc})\right) \quad \text{(A21)}$$

where the last term represents the mixing of DMS between model layers, as described for the mixing rate of DMSPd.

The first, fourth, and fifth terms in Eq. (A21) respectively represent the DMS production rate by bacterial conversion ($F^{wc}_{conversion}$), the DMS removal rates by bacterial consumption ($F^{dms.wc}_{consumption}$) and photolysis ($F^{wc}_{photolysis}$) in the water column, which are parameterized similarly to those in the bottom ice:

$$F^{wc}_{conversion} = f^{wc}_{yield} k^{wc}_{dmspd} DMSPd_{wc} \quad \text{(A22)}$$

$$F^{dms.wc}_{consumption} = k^{wc}_{dms} DMS_{wc} \quad \text{(A23)}$$

$$F^{wc}_{photolysis} = k^{wc}_{photolysis} \frac{PAR_{wc}}{PAR_{wc} + h^{wc}_{photolysis}} DMS_{wc} \quad \text{(A24)}$$

where $f^{wc}_{yield}$, $k^{wc}_{dms}$, $k^{wc}_{photolysis}$, and $h^{wc}_{photolysis}$ respectively represent the DMS yield fraction (-), the bacterial DMS consumption rate constant (d$^{-1}$), and the rate constant (d$^{-1}$) and the half-saturation constant (W m$^{-2}$) for photolysis in the water column. The reported values for $f^{wc}_{yield}$ are highly variable (0.05-1) in temperate water (Simo and Pedros-Alio, 1999) and moderately variable (0.04-0.3) in Arctic water (Luce et al., 2011; Motard-Côté et al., 2012). The only DMS yield fraction measurements available for the under-ice water column reported low values with relatively small range (0.02-0.1) as the measurements were conducted prior to the under-ice phytoplankton bloom (Galindo et al., 2015). In the standard run, $f^{wc}_{yield}$ is set to 0.2. The reported values for $k^{wc}_{dms}$ in Arctic surface water vary from 0.05 to 1.00 (mean of 0.17) d$^{-1}$ for the Canadian High Arctic in October (Luce et al., 2011) and from 0.14 and 2.2 (mean of 0.9) d$^{-1}$ for the Greenland Sea (Gali and Simo, 2010) in July. In the standard run, $k^{wc}_{dms}$ is set to 0.2 d$^{-1}$. The reported range of $k^{wc}_{photolysis}$ measured in Arctic water during the summer varies from 0.01-0.11 d$^{-1}$ for the Bering Sea (Deal et al., 2005) and the Canadian Arctic (Taalba et al., 2012) to 0.23-1.05 d$^{-1}$ for the Greenland sea (Gali and Simo, 2010). In the standard run, $k^{wc}_{photolysis}$ is set to 0.1 d$^{-1}$ and $h^{wc}_{photolysis}$ is set to 1 W m$^{-2}$.

The third term in Eq. (A21) represents the rate of change in under-ice DMS due to exchanges of DMS between the ice and water column and concentration (dilution) during sea ice growth (melting), which is parameterized in the same way as $F^{dmspd.wc}_{icesea}$:

$$F^{dms.wc}_{icesea} = \frac{1}{h_{z_0}}\left(\frac{\rho_i}{\rho_{wc}}\frac{dH_i}{dt} - \frac{\rho_{me}}{\rho_{wc}}A_{mp}r_{mp}\right)\left(DMS_{wc} - \frac{\rho_{me}}{\rho_{wc}}DMS^*\right)\delta_{z,z_0} \quad \text{(A25)}$$

where $DMS^*$ is neglected during ice growth, while it is set to DMS in the bottom ice during the flushing and melting periods.

Finally, the sixth term in Eq. (A21) represents the removal rate (nmol $L^{-1}$ $d^{-1}$) of DMS in the uppermost layer of the water column by the sea-to-air fluxes, which can be written in the vertically discretized form as:

$$F_{seaair} = f_{ow} \frac{k_{dms} DMS_{wc}}{h_{z_0}} \delta_{z,z_0} \tag{A26}$$

where $f_{ow}$ represents the fraction (-) of open water to account for fluxes through a partially ice-covered surface. In the standard run, $f_{ow}$ is set to 0 in the presence of sea ice, which is assumed to completely block the air-sea DMS fluxes. $k_{dms}$ represents the gas transfer velocity (m $s^{-1}$) for DMS. Although previous flux measurements of DMS based on the eddy covariance technique suggest that, under low to moderate winds, the gas transfer velocity can be reasonably predicted by assuming a linear wind-speed dependence (Huebert et al., 2010; Goddijn-Murphy et al., 2012; Bell et al., 2013, 2015), a recent study reconciling the eddy covariance technique with the dual tracer technique suggests that the linear wind-only-based parameterization will likely underestimate the gas transfer velocity under strong winds due to the enhancement of the bubble-mediated transfer (Goddijn-Murphy et al., 2015). In this study, the gas transfer velocity is parameterized based on Nightingale et al. (2000), which assumes a combination of linear and quadratic dependence on wind speed. Although this parameterization does not represent the bubble-mediated transfer, the gas transfer velocities predicted by this parameterization were, among other "wind speed only" parameterizations, closest to the prediction by the hybrid model of Goddijn-Murphy et al. (2015). The gas transfer velocity parameterization of Nightingale et al. (2000) was normalized to a Schmidt nubmer of 600 ($k_{600}$), and therefore, was corrected to a Schmidt number of DMS ($Sc_{dms}$) at a given temperature of ambient seawater ($T_{z_0}$ in $^\circ C$) in the uppermost layer of the water column based on Saltzman et al. (1993):

$$k_{dms} = k_{600} \left( \frac{Sc}{600} \right)^{-1/2} \tag{A27}$$

$$k_{600} = 0.333 U_{10} + 0.222 U_{10}^2 \tag{A28}$$

$$Sc_{dms} = 2674 - 147.12 T_{z_0} + 3.726 T_{z_0}^2 - 0.038 T_{z_0}^3 \tag{A29}$$

where $U_{10}$ is the observed wind speed at 10 m (m $s^{-1}$).

*Acknowledgements.* We thank the two anonymous referees for their constructive and insightful reviews of the work. This study contributes to the SCOR Working group on Biogeochemical Exchange Processes at Sea Ice Interfaces (BEPSII), the Network on Climate and Aerosols: Addressing Key Uncertainties in Remote Canadian Environments (NETCARE), and ArcticNet. NS is supported by Fisheries and Oceans Canada. AHM acknowledges support from the Natural Sciences and Engineering Research Council of Canada. The restoration data for temperature, salinity, and velocity fields in the water column were provided by Xianmin Hu (University of Alberta).

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

**Table 1.** List of the coupled sea ice-ocean sulfur cycle model variables and parameters

| Symbol | Description | Units | Value |
|---|---|---|---|
| **Variable** | | | |
| $A_{mp}$ | Melt-pond area fraction | - | |
| $DMSPp, DMSPp_{wc}$ | DMSPp concentration | nmol L$^{-1}$ | |
| $DMSPd, DMSPd_{wc}$ | DMSPd concentration | nmol L$^{-1}$ | |
| $DMS, DMS_{wc}$ | DMS concentration | nmol L$^{-1}$ | |
| $\Gamma_{z1}^{p1}$ | Grazing rate of zooplankton on phytoplankton | d$^{-1}$ | |
| $H_i$ | Sea ice thickness | m | |
| $L_{nut}, L_{nut}^{p1}, L_{nut}^{p2}$ | Nutrient limitation index | - | |
| $PAR, PAR_{wc}$ | Photosynthetically active radiation | W m$^{-2}$ | |
| $\rho_{wc}$ | Seawater density | kg m$^{-3}$ | |
| $\mu, \mu_{p1}, \mu_{p2}$ | Algal growth rate | s$^{-1}$ | |
| $T_{z_0}$ | Seawater temperature in the uppermost layer | $^{\circ}$ C | |
| $U_{10}$ | Wind speed at 10 m | m s$^{-1}$ | |
| **Parameter** | | | |
| $r_{mp}$ | Melt-pond drainage rate | m d$^{-1}$ | 0.0175 |
| $f_{active}, f_{active}^{p1}, f_{active}^{p2}$ | Active exudation fraction | - | 0.05, 0.05, 0.05 |
| $f_{sloppy}^{z1}, f_{sloppy}^{z2}$ | Sloppy feeding fraction | - | 0.3, 0.3 |
| $f_{yield}, f_{yield}^{wc}$ | Bacterial yield | - | 0.2, 0.2 |
| $h_{bi}$ | Thickness of the biologically-active bottom ice layer | m | 0.03 |
| $h_{photolysis}, h_{photolysis}^{wc}$ | Photolysis half-saturation constant | W m$^{-2}$ | 1, 1 |
| $h_{z_0}$ | Thickness of the uppermost layer of the water column | m | 1 |
| $k_{dms}, k_{dms}^{wc}$ | Bacterial DMS consumption rate constant | d$^{-1}$ | 0.2, 0.5 |
| $k_{dmspd}, k_{dmspd}^{wc}$ | Bacterial DMSPd consumption rate constant | d$^{-1}$ | 1, 5 |
| $k_{free}, k_{free}^{wc}$ | free DMSP-lyase rate constant | d$^{-1}$ | 0.02, 0.02 |
| $k_{lysis}, k_{lysis}^{p1}, k_{lysis}^{p2}$ | Cell lysis rate constant | d$^{-1}$ | 0.03, 0.03, 0.03 |
| $k_{photolysis}, k_{photolysis}^{wc}$ | Photolysis rate constant | d$^{-1}$ | 0.1, 0.1 |
| $q, q_{p1}, q_{p2}$ | Intracellular DMSP-to-Chl $a$ ratio | nmol S:$\mu$g Chl $a$ | 9.5, 100, 9.5 |
| $\rho_i$ | Sea ice density | kg m$^{-3}$ | 913 |
| $\rho_{me}$ | Sea ice density in equivalent meltwater | kg m$^{-3}$ | 1000 |

**Table 2.** Sensitivity of simulated under-ice DMS to the incorporation of the sea-ice sulfur cycle and ecosystem. Overall changes were calculated by taking the difference in the time-integrated under-ice DMS concentrations between the two runs of interest and dividing it by the time-integrated under-ice DMS concentration in the run being subtracted.

| Runs to be compared (Implications of the comparioson) | Maximum concentration difference [nmol L$^{-1}$] | Overall change [%] |
|---|---|---|
| Standard - NoIceSul (Impact of sea-ice sulfur cycle) | 2.4 | 18 |
| NoIceSul - NoIceBgc (Impact of sea-ice ecosystem) | 4.1 | -16 |
| Standard - NoIceBgc (Impact of sea-ice biogeochemistry) | 5.6 | -1 |

**Table 3.** Reported mean DMSPp:Chl $a$ and DMSPt:Chl $a$ ratios for diatom-dominated sea-ice samples.

| Ratio | Location (Arc./Ant.)[a] | Season[b] | Method[c] | Reference |
|---|---|---|---|---|
| **DMSPp:Chl $a$ (Particulate DMSP-to-chlorophyll $a$ ratio)** | | | | |
| 2.7 | Resolute Passage (Arc.) | Spring 1992 | Melting | Table 1 of Levasseur et al. (1994) |
| 1.9 | Baffin Bay (Arc.) | Sp 1998 | Melting | Bouillon et al. (2002) |
| 9.5 | Resolute Passage (Arc.) | Spring 2010 | Melting | Fig. 10a of Galindo et al. (2014) |
| 9.4 | Resolute Passage (Arc.) | Spring 2011 | Melting | Fig. 10a of Galindo et al. (2014) |
| **DMSPt:Chl $a$ (Total DMSP-to-chlorophyll $a$ ratio)** | | | | |
| 8.4[d] | Weddell Sea (Ant.) | Spring 1988 | Melting | Table 1 of Kirst et al. (1991) |
| 22[e] | Southern Ocean (Ant.) | Winter-Spring 1997 | Melting | Trevena et al. (2000) |
| 37[e] | Prydz Bay (Ant.) | Spring 1997-1998 | Melting | Trevena et al. (2003) |
| 20 | Barrow (Arc.) | Winter-Spring 2002 | Melting | Uzuka (2003) |
| 49 | Weddell Sea (Ant.) | Spring 2004 | Dry-crushing | Fig. 6a of Tison et al. (2010) |

[a] Arc.: Arctic; Ant.: Antarctica

[b] Winter: Jan.-Mar. (Jul.-Sep.) for Northern (Southern) Hemisphere; Spring: Apr.-June. (Oct.-Dec.) for Northern (Southern) Hemisphere

[c] Either melting or dry-crushing as described by Stefels et al. (2012).

[d] Average of brown ice and ice core samples.

[e] Calculated based on mean DMSPt and Chl $a$ values given in Table 6 of Trevena et al. (2003).

**Table 4.** Sensitivity of simulated bottom-ice and under-ice DMS to doubling the model parameter of the sea-ice sulfur cycle. Changes in the bottom-ice and under-ice DMS were calculated by subtracting the time-integrated DMS in the sensitivity run from the time-integrated DMS in the standard run and dividing the difference by the time-integrated DMS in the standard run.

| Run | Description | Change in the bottom-ice DMS | Change in the under-ice DMS |
|---|---|---|---|
| Case 1 | Doubling the intracellular DMSP:Chl $a$ ratio | 100 % | 17 % |
| Case 2 | Doubling the DMS yield fraction | 91 % | 12 % |
| Case 3 | Doubling the bacterial DMSPd consumption rate constant | -2 % | -2 % |
| Case 4 | Doubling the bacterial DMS consumption rate constant | -44 % | -5 % |
| Case 5 | Doubling the photolysis rate constant | -8 % | -2 % |

**Table 5.** Sensitivity of simulated sea-air DMS fluxes to the open-water fraction and to the incorporation of sea-ice biogeochemistry. Overall changes were calculated by taking the difference between the two runs of interest and dividing it by the cumulative flux in the subtracted run.

| Open-water fraction [-] | 0.02 (Small lead) | 0.1 (Large lead) | 0.5 (Near ice margin) | 1 (At ice margin) |
|---|---|---|---|---|
| Maximum flux [$\mu$mol m$^{-2}$ d$^{-1}$] | | | | |
| (Standard) | 0.3 | 1.2 | 4.9 | 8.1 |
| Maximum flux difference [$\mu$mol m$^{-2}$ d$^{-1}$] | | | | |
| (Standard - NoIceSul) | 0.1 | 0.2 | 1.1 | 1.7 |
| (NoIceSul - NoIceBgc) | 0.1 | 0.5 | 2.1 | 3.5 |
| (Standard - NoIceBgc) | 0.1 | 0.7 | 3.0 | 5.0 |
| Overall change [%] | | | | |
| (Standard - NoIceSul) | 20 | 24 | 26 | 26 |
| (NoIceSul - NoIceBgc) | -14 | -13 | -11 | -9 |
| (Standard - NoIceBgc) | 3 | 8 | 12 | 15 |

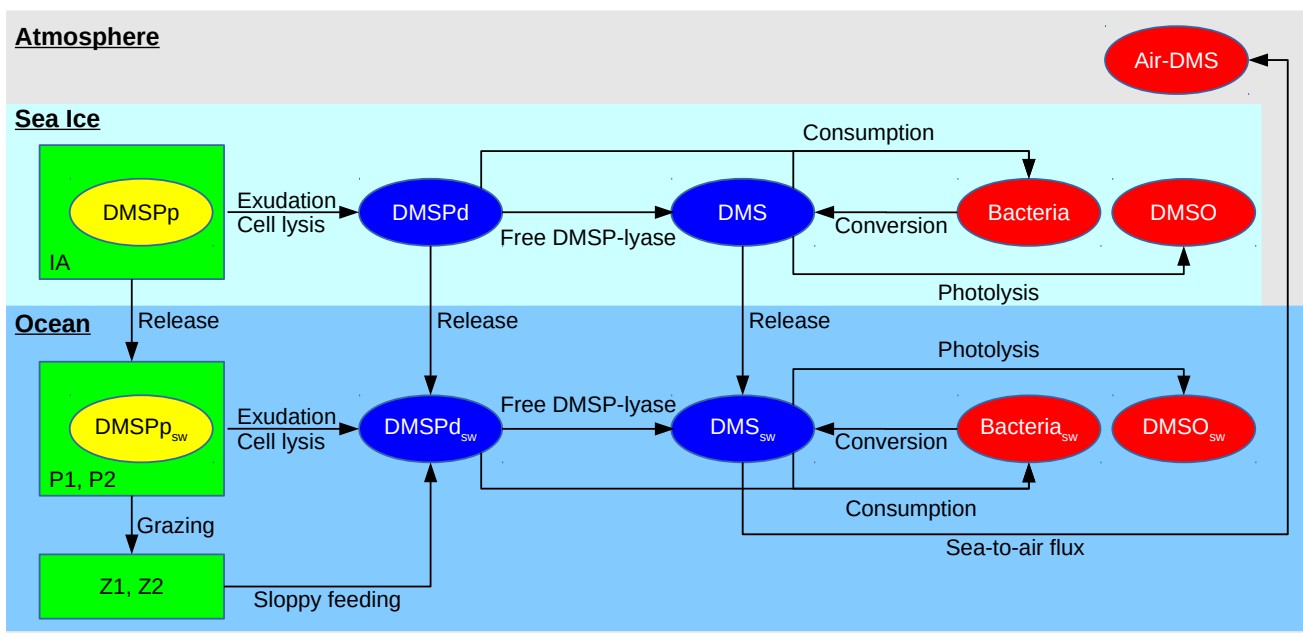

**Figure 1.** Schematic of the sea-ice and oceanic components of the sulfur cycle module. Variables in blue (yellow) are simulated prognostically (diagnostically), while the variables in red are not simulated but the relevant processes are parameterized. Variables in green are simulated prognostically by the ecosystem model. Arrows represent the physical and biogeochemical fluxes parameterized in the module.

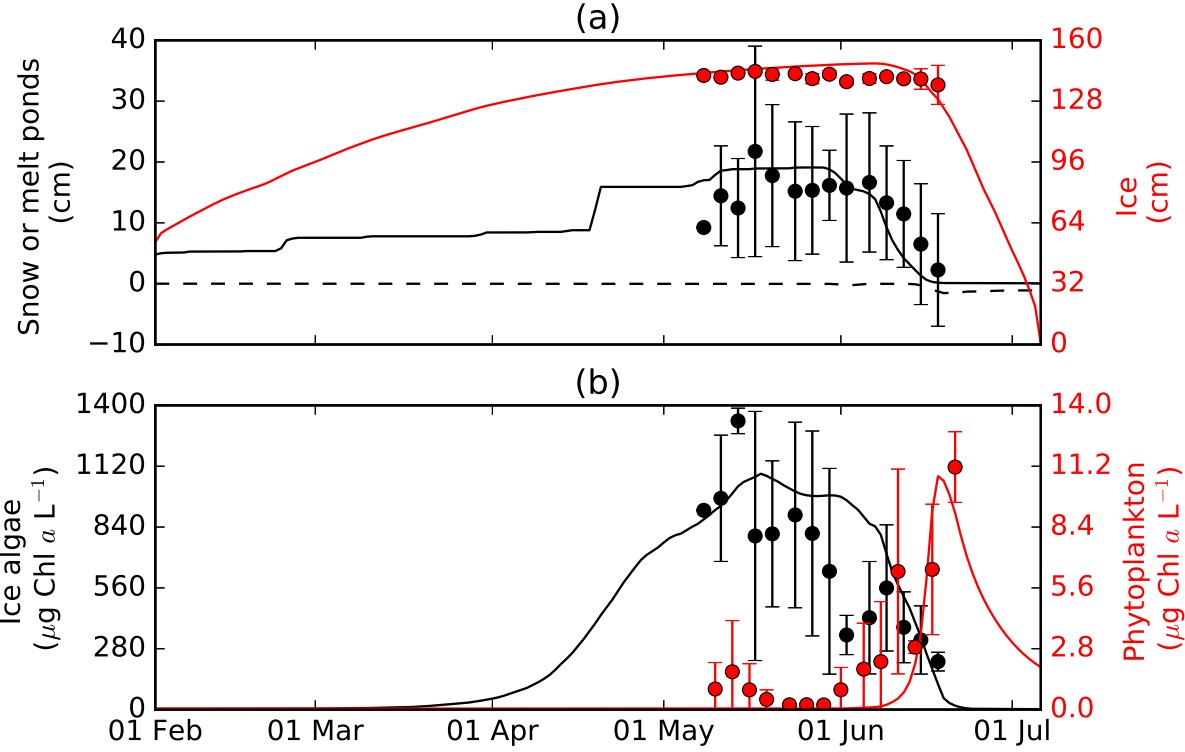

**Figure 2.** Simulated (lines) and observed (dots and bars) time series of (a) snow (black solid) and melt pond (black dashed) depths [cm] and ice thickness [cm] (red), and (b) ice algal biomass [$\mu$g Chl $a$ $l^{-1}$] in the bottom 3 cm ice (black) and phytoplankton biomass [$\mu$g Chl $a$ $l^{-1}$] averaged over the upper 10 m water column (red) in Resolute Passage during 2010. In (a), the negative values present the depth of melt ponds. Also in (a), the observed values show the average (dots) and 1 standard deviation (vertical bars) of samples collected at three sites of high (>20 cm), medium (10-20 cm), and low (<10 cm) snow cover. In (b), the observed ice algal biomass shows the average (black dots) and 1 standard deviation (vertical bars) of samples collected in ice cores under high, medium, and low snow cover sites, while the observed phytoplankton biomass shows the average (red dots) with $\pm$ 1 standard deviation (vertical bars) of samples collected in seawater at 1.5, 2, 5, and 10 m depth. Note that the biomass for both ice algae and phytoplankton is expressed in terms of volumetric concentration. Hence, despite high concentrations in the sea ice, they are confined to a very small vertical range (3 cm) compared to those concentrations in the upper 10 m of the water column.

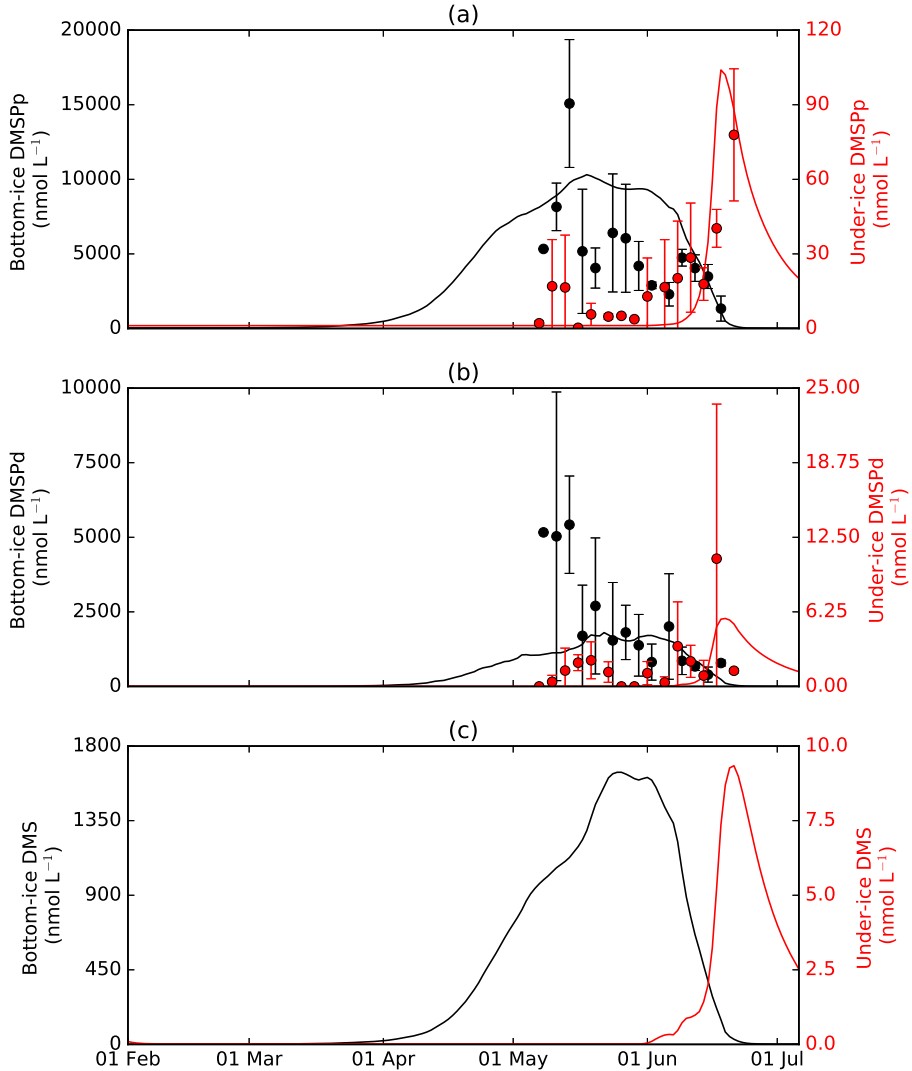

**Figure 3.** Simulated (lines) and observed (dots and bars) time series of (a) DMSPp, (b) DMSPd, and (c) DMS concentrations [nmol L$^{-1}$] in the bottom 3 cm ice (black) and averaged over the upper 10 m water column (red) in Resolute Passage during 2010. The observed bottom-ice values show the average (black dots) and 1 standard deviation (vertical bars) of samples collected in ice cores under high, medium, and low snow cover sites. The observed upper 10 m water column values show the average (red dots) with $\pm$ 1 standard deviation (vertical bars) of samples collected in seawater at 1.5, 2, 5, and 10 m depth.

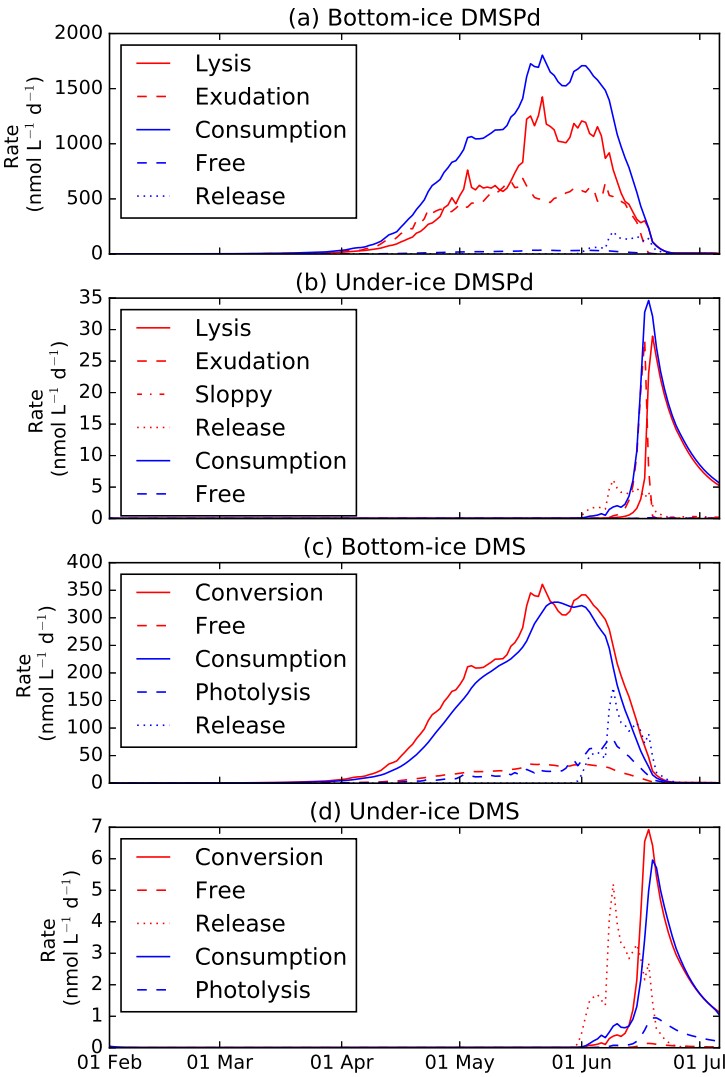

**Figure 4.** Simulated time series of daily mean production (red) and removal (blue) rates [nmol L$^{-1}$ d$^{-1}$] of (a and b) DMSPd and (c and d) DMS (a and c) in the bottom 3 cm ice and (b and d) in the uppermost layer (0.5 m below the ice) of the water column. In (a) and (b), the sources for DMSPd are cell lysis (Lysis; solid red), exudation (Exudation; dashed red), and sloppy feeding (Sloppy; dash-dot red in (b) only) while its sinks are bacterial DMSPd consumption (Consumption; solid blue) and free DMSP-lyase (Free; dashed blue). In (c) and (d), the sources for DMS are bacterial DMSPd-to-DMS conversion (Conversion; solid red) and free DMSP-lyase (Free; dashed red), while its sinks are bacterial DMS consumption (Consumption; solid blue) and photolysis (Photolysis; dashed blue). Release from the bottom ice (Release; dotted) is a sink for the bottom-ice DMSPd (a) and DMS (c), while it is a source for the under-ice DMSPd (b) and DMS (d).

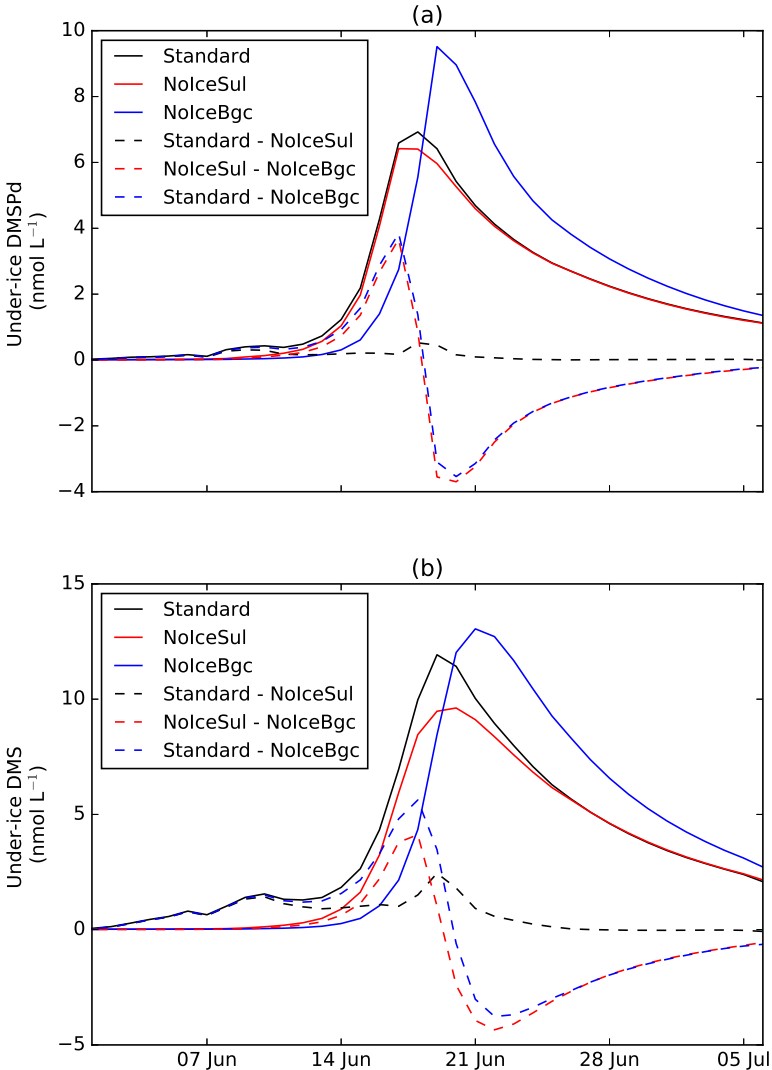

**Figure 5.** Simulated time series of (a) DMSPd and (b) DMS concentrations [nmol L$^{-1}$] in the uppermost layer (0.5 m below the ice) of the water column during the melt period in 2010 for the standard run (Standard) and the sensitivity runs that exluded the sea-ice sulfur cycle (NoIceSul) and both the sea-ice sulfur cycle and ecosystem (NoIceBgc). Dashed lines represent the concentration difference between the two runs of interest. Positive differences represent enhancement in the concentration due to the incorporation of sea-ice sulfur cycle (Standard - NoIceSul), sea-ice ecosystem (NoIceBgc - NoIceSul), and both sea-ice sulfur cycle and ecosystem (Standard - NoIceBgc), respectively, while negative values represent reduction.

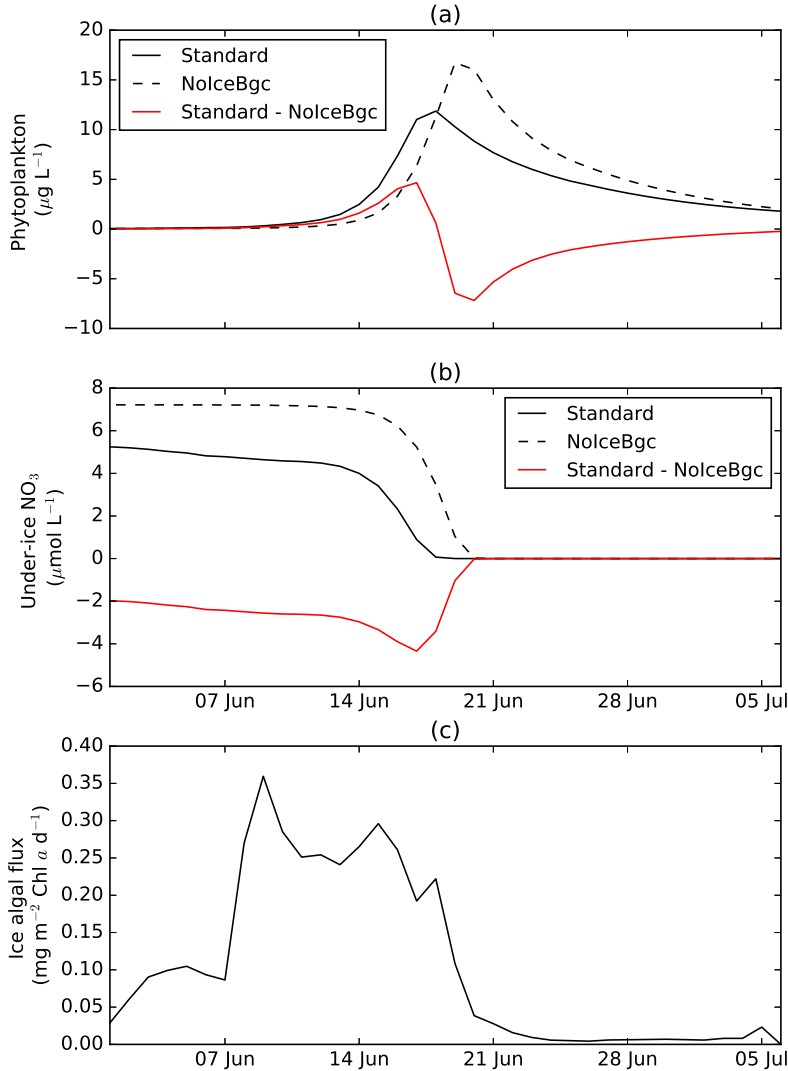

**Figure 6.** Simulated time series of (a) phytoplankton biomass [$\mu$g Chl $a$ L$^{-1}$] and (b) nitrate concentration [$\mu$mol L$^{-1}$] in the uppermost layer (0.5 m below the ice) of the water column during the melt period in 2010 for the standard run (Standard) and the sensitivity run that exluded both the sea-ice sulfur cycle and ecosystem (NoIceBgc). (c) Ice algal flux [mg Chl $a$ m$^{-2}$ d$^{-1}$] entering the large phytoplankton pool in the uppermost layer of the water column. In (a) and (b), red lines represent the respective differences in phytoplankton biomass and nitrate concentration between the standard and sensitivity runs.

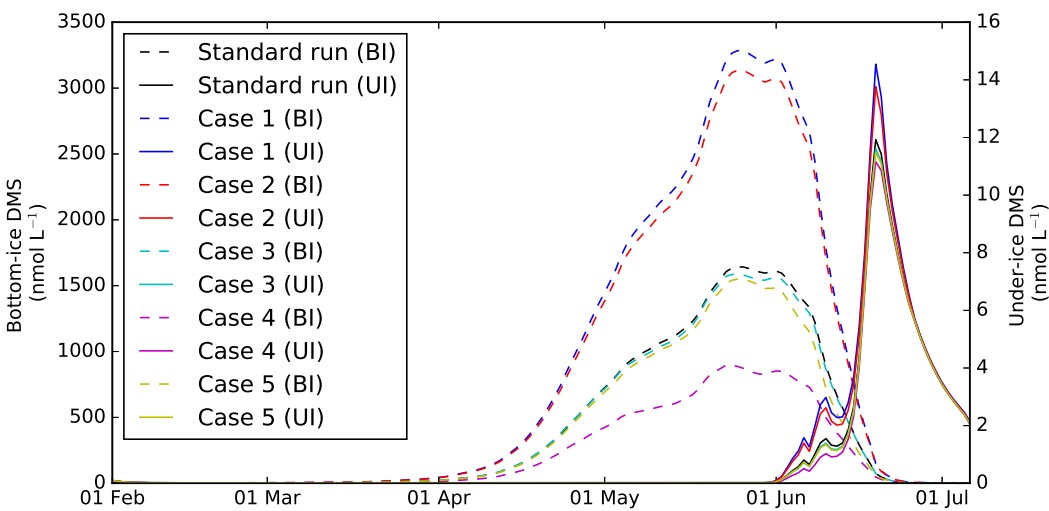

**Figure 7.** Simulated time series of bottom-ice (BI) and under-ice (UI; 0.5 m below the ice) DMS concentrations [nmol L$^{-1}$] during 2010 for: the standard run; Case 1: doubling the intracellular DMSP:Chl $a$ ratio; Case 2: doubling the DMS yield fraction; Case 3: doubling the bacterial DMSPd consumption rate constant; Case 4: doubling the bacterial DMS consumption rate constant; and Case 5: doubling the photolysis rate constant.

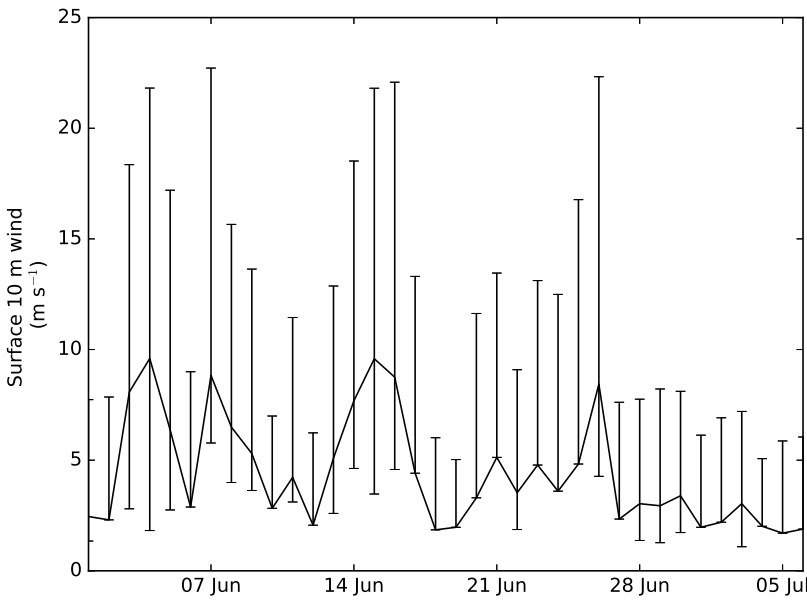

**Figure 8.** Time series of daily mean surface 10-m wind speed [m s$^{-1}$] observed at the Resolute airport (located within 7 km of the study site) during the melt period in 2010. The upper and lower vertical bars associated with the daily mean values represent the daily maximum and minimum values, respectively.

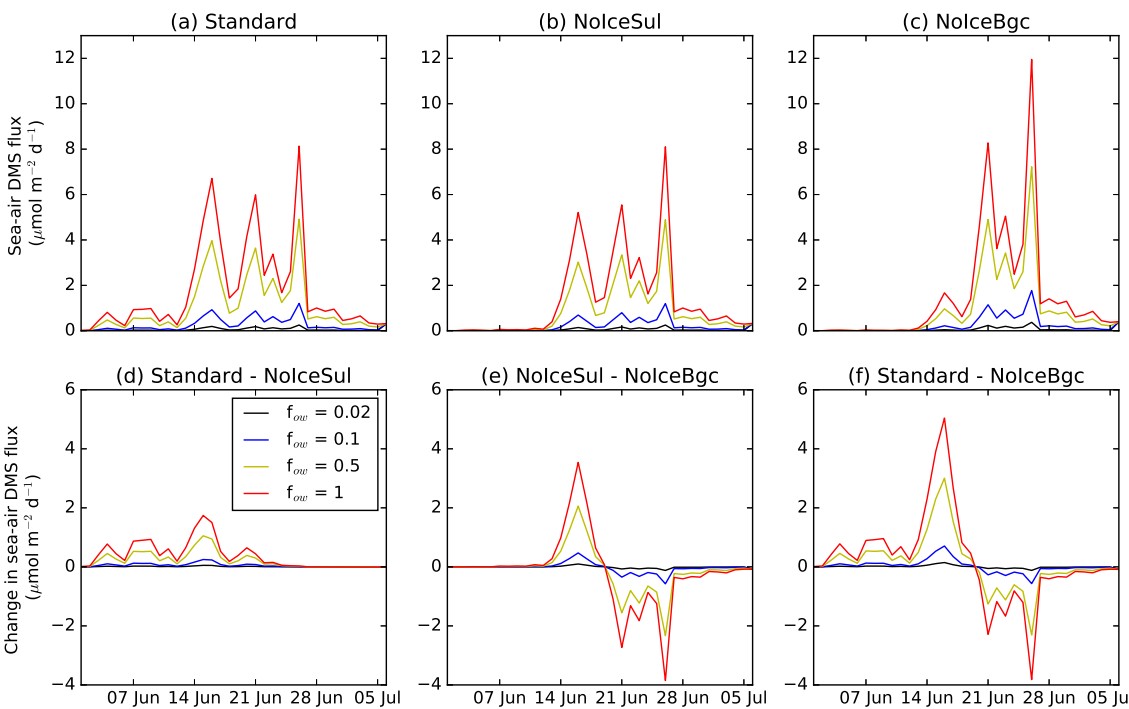

**Figure 9.** Simulated time series of sea-air DMS fluxes [$\mu$mol m$^{-2}$ d$^{-1}$] for (a) the standard run and the sensitivity runs that excluded (b) the sea-ice sulfur cycle (NoIceSul) and (c) both the sea-ice sulfur cycle and ecosystem (NoIceBgc) and the flux difference between (d) the standard and NoIceSul runs, (e) the NoIceSul and NoIceBgc runs, and (f) the standard and NoIceBgc runs during the melt period in 2010. In (d), (e), and (f), positive values represent enhancement in the simulated flux due to the incorporation of sea-ice sulfur cycle, ecosystem, and both sulfur cycle and ecosystem, respectively, while negative values represent reduction.

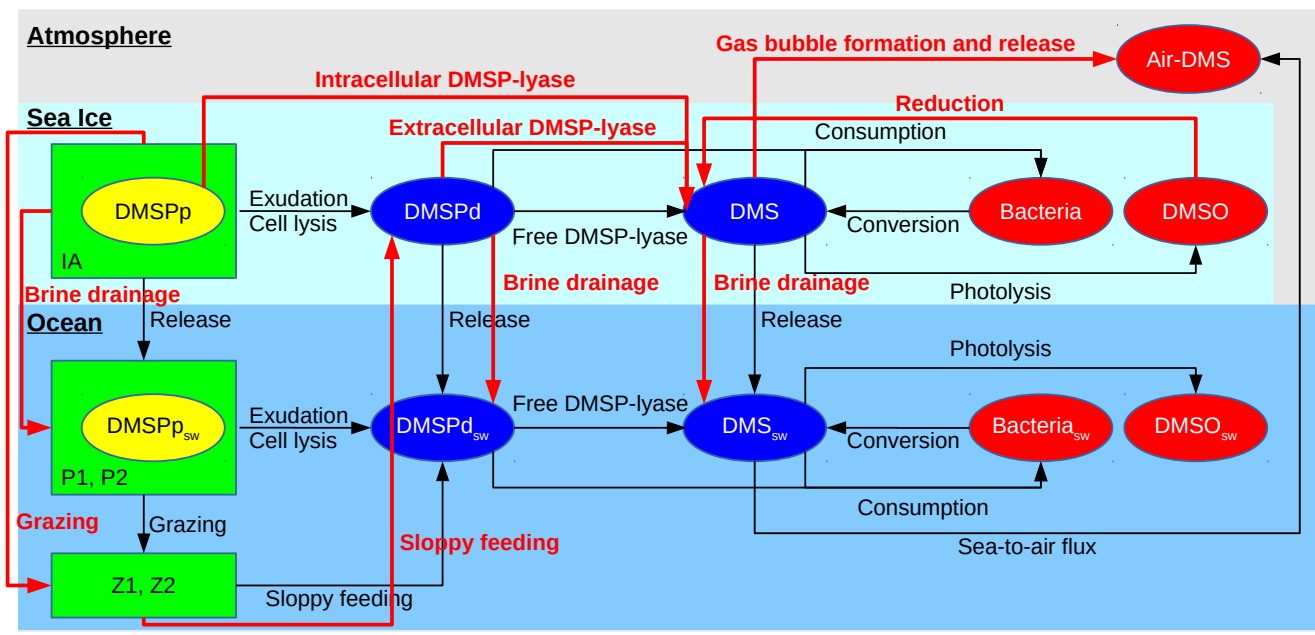

**Figure 10.** Same as Fig. 1, but with additional physical and biogeochemical fluxes suggested for future model development (red arrows).