# Peer review of "Implications of sea-ice biogeochemistry for oceanic production and emissions of dimethylsulfide in the Arctic"

_Biogeosciences, 2016_

## Referee Comment (RC1) · Anonymous Referee #1 · 30 Nov 2016

Hayashida et al propose a model study of the sulphur cycle in the Arctic landfast ice zone.

The quality of the text and figures is quite good. The authors have the interesting conclusion that sea ice sulphur cycle and ecosystems have considerable impacts on DMS production under the ice and should be considered in estimates of ocean DMS fluxes.

Whereas I believe this conclusion is potentially supported by the scientific elements of the paper, the current presentation did not convince me.

1) I'm not sure that sulphur is conserved in the model. This question is central: if

the authors suggest an enhancement of sulphur fluxes to the atmosphere, they must explain where this extra sulphur comes from. I don't see in the text or in the figures which reservoir is loosing sulphur in your model.

2) I cannot understand the mechanisms in the sensitivity experiments from the text. Section 3.2.1, they authors explain they turn the sulfur cycle off. Which terms of the equations does that represent ? What happens with DMS in these experiments ? What is the chain of mechanisms leading to the decrease in under-ice DMSPd and DMS and decreasing the air-sea fluxes ?

This is my key criticism. If the authors can at least highlight how sulphur is conserved among the different sulphur forms and, more importantly, explain more in depth the mechanisms in their sensitivity experiments, this can make a good paper.

—

A few more detailed comments

- Model description is not convincing. This could probably be fixed by better explanations (not more of them).

1) Physical and ice algal components models come from a paper under review. Next time the authors should consider to attach the companion paper.

2) I doubt of sulphur conservation. I don't clearly see how sulphur can be conserved now. It should be visible from the equations. For instance, I don't see where the losses of DMSPp in sea ice go (eq A1). Make sure sulphur conservation is obvious from the few evolution equations.

3) Some physical terms (notably sea ice growth and melt for the first ocean layer) are completely absent from the equations, which is surprising, because these are leading-order terms for most other biogeochemical compounds.

4) Is there any good reason not to use a standard formulation of the DMS air-sea flux

? In your section 3.2.3, they are just proportional to DMS_water. As far as I know, this is not in line with classical air-sea flux formulations. It would not be hard to introduce solubility and pDMS in the atmosphere.

- Sensitivity experiments are difficult to understand. I have trouble to distinguish between "sea ice ecosystem" and "sea ice sulphur cycle" sensitivity experiments, because the sulphur cycle is partly controlled by sea ice algae. So the authors should clearly tell which terms are involved and, most importantly explain the mechanisms involved.

- In figures, the authors often compare model concentrations in $\mu$mol/L to observed ones. This could introduce a source of bias is the depth of the extracted core section does not match the 3 cm of the model. I would suggest to rescale observations to 3 cm if possible.

- Some aspects of the intro (links between DMS and cloud nucleation) may not be in phase with literature. I felt the role of DMS was a little bit overstated. Line 10 of page 2, the authors point DMS as the driver of arctic clouds backed with one unique citation. I was surprised by this statement.

I had a little trip in the literature, and discovered that this should probably be nuanced. Tjernström et al (ACP 2014) mention page 5 that they are indeed looking for a missing source of aerosols in the Arctic. Yet they did not found H2SO4 but rather organic molecules polymer saccharide molecules. Later on, page 2828, they explain that "This suggests a stronger possible link between marine biology, cloud prop- erties and climate than provided by DMS alone (Leck and Bigg, 2007)".

Tjernström et al. Atmospheric Chemistry and Physics, 14, 2823-2869, 2014.

I'm sure the authors are aware of these works, and I would here just suggest to better explain the the links between DMS and Arctic clouds, even if it takes a few sentences.

- The introduction felt generally a little bit "inbred", with lots of references coming from

the own group of the authors. If there is no other choice, skip this comment.

---

## Referee Comment (RC2) · Anonymous Referee #2 · 21 Dec 2016

General comments: The study of Hayashida et al. introduces a new coupled sea ice-ocean ecosystem-sulfur cycle model. The purpose of the model is to determine the DMS,P production of Arctic sea ice microbial communities and its contribution to oceanic emissions of DMS in the Arctic. This contribution is important to assess since: -Oceanic DMS emissions could play a paramount role in aerosols formation in the relative pristine atmosphere of the Arctic Ocean. Hence, DMS could be a key compound regulating the regional climate. -Sea ice microbial communities generally show very high production of DMS,P, with concentrations several orders of magnitude higher than typical oceanic concentrations. -The sea ice cover in the Arctic is changing rapidly in response to regional warming, generating potential important feedback loops. In my

opinion, the general context of the work of Hayashida et al. is therefore highly relevant.

As correctly introduced by the authors, both the DMS,P production of sympagic communities and its contribution to oceanic emissions of DMS are both unfortunately currently poorly constrained. Only a handful of studies have determined DMS,P (DMSPd, and DMSPp mainly, only one or two studies have reported DMS concentrations) concentrations in Arctic sea ice, and only a couple have tackled production and removal processes and rates. Close to nothing is known about the transfer of DMS,P at the ice/ocean/atmosphere interfaces. In addition, sea ice typically shows a very high temporal and spatial variability making harsh any extrapolation of field studies that are limited in space and time. In this context, I am really happy to see the modelling effort developed by Hayashida et al. As detailed in my minor comments, it brings both interesting answers and questions to the sea ice DMS community and is therefore worth publishing.

As it is the first time such a perilous (given the complexity of the sea ice DMS cycle) enterprise is undertaken, there are however a lot of caveats and limitations that I think the authors could do a better job at presenting in the paper. This is in fact my major comment or request to the authors. The caveats and limitations are only partially tackled in the discussion and in the conclusion. I think they deserve their own section in the discussion part of the manuscript. I would like this section to: 1) Present the caveats and limitations (see my minor comments) of the model itself, in particular the fact that brine dynamics are neglected. 2) Present recommendations for observational studies, something like "what modellers need from field studies".

This section could also tackle the second important weakness of the paper, the fact that the model is validated by only one time series study that does not even cover all the outputs of the model. Is there really not any other data available in the literature that you could have use to validate the model or at least to give an idea to the reader on the applicability of your model to other locations/ice conditions in the Arctic? The limited time frame of the Galindo et al data set which partially miss two important phases of the

cycle presented by the model does not help to build confidence. I think that throughout the manuscript you could better put your results and parameters in perspective with the literature, even if you have to use the more abundant Antarctic sea ice DMS literature. I know that most of the drivers are significantly different in the Antarctic, but some basic physical and biological concepts are the same, especially if you look at fast ice studies with no surface flooding. It is also really important in this section, or in other sections of the paper, to better convince the reader that your model is applicable to other parts of the Arctic and not only to the study site of Galindo et al. A way to do this would be to better describe the study site and conditions of Galindo et al and to show that they correspond to other sites/and conditions in the Arctic.

One final major concern that I have reading the paper is the use of the term flushing throughout the manuscript. It is never described in the manuscript and I am having trouble knowing what the authors really refer to. Flooding is a very specific process defined as: "Flushing refers to the washing out of salty brine by relatively fresh surface melt water that percolates into the pore space during summer". I have the feeling that flushing is used as another term for "release" by the authors and could in fact include other form of material release from sea ice (e.g. brine drainage, melting of bottom ice). Could you please better define this in the manuscript and replace by another term if necessary? In addition to these general comments, I also have additional minor comments and suggestions listed here after:

Specific comments: • Abstract Page 1: Line 5: Please use the plural form: The result"s" of the 1-D model "were" compared. Also consider using plural for DMS emission"s". Line 6: Please use the plural form: our result"s" reproduced. Line 7: Flushing is a very distinct process in sea ice thermodynamics. Please consider using the word "release" more generic than flushing, unless you are exclusively talking about flushing. This comment actually applies to the whole document. Line 9: "Processes that dominated the budgets of bottom-and under-ice. . ." This sentence is a little bit heavy, consider rephrasing. Line 15: "would be better constrained by new observations".

The whole sentence sounds like a recommendation, consider using "should" instead of "would".

• Introduction Page 1: Line 18: "is a volatile biogenic trace gas" A gas is always volatile, please use either volatile or gas in the sentence. Line 19: "planktonic and microbial" Microbe relates to the size of organisms, and so includes some planktonic organisms. The use of the two words is a little bit confusing here. Line 20: "Oceanic DMS emissions also play. . ." I would say "can play" especially considering your next paragraph about the controversy on the global scale effect of DMS.

Page 2: Line 7,8: This sentence is very long, consider splitting it in two. Line 8: "increase and peak" I would say: "were observed to increase and peak". Line 16: ""high concentrations at the bottom" I would say "in the bottom". Line 19: I would stick only to Arctic studies here, so please remove Kirst et al., 1991 from the reference list, or clearly indicate that the study of Kirst et al. 1991 covers the Antarctic. Line 26: I would like to see one or two sentences saying what the outcome of the model study of Elliott et al. was and how your study differs from their approach. I think it is important to frame your study. Line 27: Please consider starting the paragraph with "In this study, we test the hypothesis that. . . by developing a sulfur cycle. . .". Line 30: "The rest of this study. . ." I don't think this part is necessary unless required by the journal.

• Model description and experimental design Page 3: Line 14: "diffusive exchange of nutrients at the ice-water interface". Does (and if yes how) the model consider the role of brine convection in the exchange of nutrients at the ice-water interface? Several authors and Antarctic based models have highlighted the importance of this process, see Vancoppenolle et al. 2010 for instance ("Modeling brine and nutrient dynamics in Antarctic sea ice: the case of dissolved silica"). Line 23: Please add "emphasized that" "the" sulfur cycle represented. . .". Line 32: It should be clearly stated here that cell lysis and exudation rates are taken from limited water column measurements and not sea ice/brine measurements.

Page 4: Line 4 to 7: This sentence is hard to understand, please rephrase or make 2 separated sentences. Same comment as in the abstract regarding the term "Flushing". Line 10: Good, I like to read this, but not only for S but also for nutrients, see my previous comment. Line 14, 15: Ignoring sloppy feeding is unfortunately a weakness of the paper given the relative importance of grazing on bottom ice communities. Same for the DMSO reduction to DMS. Model setup: perhaps some general information about the Arctic-ICE Resolute passage sampling area could be provided to the reader here. Typical ice thickness developing, type of ice (thickness, texture), typical date of formation and retreat, ice deformation, typical ice and weather regimes. . . I think that somewhere you need to convince the reader that your site is somewhat representative of a more general situation e.g. Arctic coastal fast ice. Line 27: What is the total water depth at the Arctic-ICE sampling site? Line 31: Was a met tower deployed at the sampling site? If yes, why not using data from the tower? Or at least compare the two data sets. Line 32: I would like to see in addition to Dukhovskoy et al. 2016 a more general reference for the NEMO-LIM2 model. Line 34: Where the initial ice thickness, snow thickness, and melt-pond depths set based on field observations? This is not clear.

Page 5: Line 1: "to match with observations" To match with observations of the real depth of the skeletal layer, or to match with the vertical sampling resolution used by Galindo et al.? I think this should be clarified. Line 2: It is not clear from your writing if the initial biomass of ice algae was set based on the measurements of Mundy et al. (2014) as for the nutrients. Same comment for ammonium and particulate silica. Line 7: This is a strong assumption given that your initial chl a is set to 3.5 $\mu$g.L-1. In oceanic waters, for such chla concentrations you already have non negligible DMS and DMSPd concentrations. And it is definitely the case for sea ice as well, even in diatom dominated communities (see several Antarctic studies for instance, e.g. Carnat et al., 2014 "Physical and biological controls on DMS,P dynamics in ice-shelf influenced fast ice during a winter-spring and a spring-summer transitions" JGR.). How do you justify this? Line 14: Are these "field measurements in Arctic waters" referenced somewhere in the manuscript? Line 15: "Emissions"s".

[Figure]

 c Results and discussions Page 5: Line 30: How did you deal with melting and refreezing of the snow pack (typical in early summer in the Arctic) and the formation of superimposed ice? Was superimposed ice detected in the field samples? Was superimposed ice considered as snow or ice in the obs?

Page 6: Line 9: Yes, and this is why it is important to give general information about the sampling site to the reader. Is this assumption based on observations during the 2010 campaign or during other years? Do you have a reference for this? Line 24: What about the increase in observed under-ice chla in mid-May and the localized increase in early-June? Could they be the sign of brine drainage not simulated by the model? I think the hypothesis has to be written somewhere. You see the same trend in DMSPp and to a lesser extent in DMSPd. Line 27: It would be nice to indicate here the fixed DMSP:chla ratio used and not only refer to the equations. Line 28: How strong was the relationship? Please indicate this. Line 32: How do you explain the spatial variability? Spacing of brine channels? Patchiness of ice algae in bottom ice? Hints should be given here.

Page 7: Line 9 and 10: "To the release of ice algae from bottom ice". Through which process? Bottom sea ice melting? Hanging algae that just drop in the water? Brine drainage? Flushing? Line 13 and 14: The DMSPp sinking part does not convince me at all. What is the argument to support this? Just that the DMSPp peak rapidly disappears during the next sampling event? Do you have DMSPp measurements in deeper ocean layers during the same sampling event? It could be also that most of the DMSPp is rapidly converted to DMSPd and DMS. You do not have observations of DMS, so you cannot rule that second option out. I think that you should mention both options and not only focus on the sinking of DMPSp. Line 23 to 25: The explanation given for the mismatch between the three first observations and the model simulation in DMSPd does not convince me at all for the moment. We need to know at which snow depth day 1 was sampled and what the effect could be on DMSPd. Why would spatial variability be higher during day 2 than during any other sampling day? Is the fact that all the

three observed variables (DMSPp, DMSPd, chla) were significantly higher than model predictions during day 3 not at sign that the model is missing something? What about other ancillary parameters observed during the 3 first sampling days? (community composition for instance?, stress on the algae?). Line 31: "Degradation of DMSPp to DMSPd" that you disregarded before, saying that DMSPp was sinking...see line 13 and 14.

Page 8: Line 3: "upper 10 m" perhaps add "of the water column". Line 7: Considering the comment you make Line 28-30 of page 11 on the cell lysis and artificial DMSP release (and hence DMSP conversion to DMS) on melting of ice samples, how can we trust the 2000 nmol.L-1 value given in Levasseur 2013. How was the sea ice DMS measured in that study? Section 3.1.4.: I know there is no sea ice field observations to validate these various production and removal rates but it would be nice to at least compare them to oceanic field observations so that the reader can evaluate if the numbers make sense.

Page 9: Line 1: "due to low zooplankton biomass during the melt period" Is this supported by any kind of observation? Line 21-22: "In mid-June, the simulated DMS..." This sentence is hard to understand, please rephrase or cut in two separated sentences. Line 25-30: This whole part is a little bit hard to follow. It is hard to know when you are talking about pre bloom conditions, early bloom conditions, peak bloom conditions...I would suggest rewriting like this (line 25): "On the other hand, during the initiation of the under-ice bloom, the simulated DMS production rates by flushing were comparable to the rates of other simulated processes associated with the bloom, reaching a maximum value (>5 nmolL.d) in early June. "

Page 10: Line 1-2: What about the upward vertical migration of DMS within the brine channels? It is very likely that in an open system such as warm interconnected brine channels, gaseous compounds can migrate upwards due to buoyancy if present under the form of bubbles. Would DMS form bubbles at some point? See Zhou et al., 2013 "Physical and biogeochemical properties in landfast sea ice (Barrow, Alaska): Insights

on brine and gas dynamics across seasons", or Moreau et al. 2014 "Modelling argon dynamics in first-year sea ice". Did you consider this? If not, why? I think this needs some clarification. Line 23: The use of the word "significantly" suggests that statistical tests were conducted. Is it the case? And if yes, please indicate the level of significance. If no, please use another term. Line 24-25: Please rephrase: "In comparison to the NoIceSul run, the NoIceBgc run simulated under-ice DMSPd and DMS concentrations that were higher in late June onward." Line 28: "Heightened"... "higher"? Line 27-28: This needs further explanation. How does the absence of ice algae increase the nutrients available to under-ice algae? It is not clear from you text how they get access to the same nutrient pool? Is it the fact that the nutrients not consumed in bottom ice are "flushed" to the under-ice water ? Or the fact that the under-ice nutrient pool is somehow consumed by bottom ice algae? If yes, how? Also this whole section lacks comparisons with observations of nutrient/algae dynamics from the literature. It would also be nice here to remind here that the effect of brine convection on nutrient dynamics (see Vancoppenolle work) was not taken into account in the model. Line 34: This sentence is too long and complex, please split in two or simplify.

Page 11: Line 4: of "the" sea ice ecosystem. Line 10: Remove one of the "the". Again, you need to explain to the reader how the dynamics of the bloom were precisely affected. Whole section: What about the self-shading effect? Removing bottom ice algae should increase the amount of light available to under-ice algae. Did you consider this? Line 10: Influencing the dynamic of the bloom...through the control exerted on nutrients? How? Line 22-25: This sentence is too long, please split in two. Line 25-30 and line 28-30: You should clearly state here which study were conducted in the Arctic and which were conducted in Antarctica. Also, you need to discuss major algal groups in the assemblages sampled by the different studies. Line 28-30: Right, I agree with this. Cell rupture on melting of the samples will cause artificial release of DMSPp and transformation into DMSPd and DMS in contact with free or cell bound enzymes. But then, how can you trust the DMSPp and DMSPd measurements of Galindo et al that you use to validate the model?? Furthermore, how can you trust the DMS measurement from

[Figure]

Levasseur 2013? All these measurements were made by melting sea ice samples. There is at least one study in the literature that reports DMSP:chla ratios using other techniques than melting, you should refer to it. For example, Carnat et al. 2014 "Physical and biological controls on DMS,P dynamics in ice-shelf influenced fast ice during a winter-spring and a spring-summer transitions" JGR.) gives DMSPt:chla ratios for bottom ice diatom dominated communities. I think the best way to present this whole DMSP:chla issue is to draw a quick table with a few references from the literature indicating DMSP(p or t):chla ratios measured, community compositions, arctic/Antarctic, melting or other extraction technique. This table could go in the supplementary but would be a very nice addition to the paper.

Page 12: Line 5-6: It would be nice here to restate quickly which parameters then control the temporal patterns of DMS concentrations. Line 14-15-16: This is very nice to read. Line 23-24: I am not sure to understand this correctly. Is this because DMSPd removed by bacterial consumption is not available anymore to free DMSPd lyase? Is there because part of the DMSPd pool used by bacteria is converted partially in other compounds than DMS? Line 25: Ok, but which other parameters would you recommend observational studies to target?

Page 13: Line 15: How did you deal with fetch of small leads and even large leads compare to open ocean parametrizations that you use? Also, how did you deal with shear-driven and convection-driven turbulence in the sea ice zone that have been shown to have a huge impact on k (see Loose et al., 2014; A parameter model of gas exchange for the seasonal sea ice zone.) I do not especially ask you to redo all the math with a new or corrected parameterization, but at least the fact that these important parameters were neglected should be mentioned. Line 16: Again, was any met tower deployed on site? Line 29-30: Pandis and Russel 1994 are old papers, is there not more recent work on particle formation associated with DMS flux? Line 30-31: "saturation" is an odd term to use here. I think the concept could be better explained with one or two additional sentences.

Page 14: Line 16: Occurrence instead of occurrance.

Page 15: Line 1: A reference for the summertime flux for ice free waters in the Arctic is missing here. I don't know however if the journal allows ref in the conclusion. Line 2 to 4: I totally agree with this part but I believe it should be a distinct discussion section of the paper rather than a few lines in the conclusion as mentioned in the general comments. Line 13: I am a little bit surprised that nowhere in your recommendations/conclusions you are talking about the need for additional DMS,DMSP, DMSO data to validate models. I am also surprised that you do not say a word about brine drainage, although I believe it is a key missing item in your paper.

Figure 1: Same comment as in the abstract regarding the term "Flushing". Figure 3: The use of the different vertical scales for DMSPp, DMSPd, and DMS is slightly confusing. Is there a way to use similar vertical scales for the three compounds (by adding breaks or log scale?) so that they could be better compared? Figure 4: Please check the legend. In the version that I have, the dashed lines do not show up. Also, in all graphs, please use parenthesis instead of the concentration symbol in the axis titles. Figure 5: Same comment as for Figure 5, dashed lines do not show up in the legend box. Figure 8: Perhaps indicate in the figure caption the distance between the sampling site and the Resolute Airport. Other figures are fine.

---

## Author Comment (AC1) · 3 Mar 2017

**We, the authors, thank Anonymous Referee #1 for the thorough and constructive review of our manuscript. We revised the manuscript by accommodating the referee's feedback as much as possible. In the following, we provide our responses (written in red) to the referee's comments (written in black).**

Anonymous Referee #1

Hayashida et al propose a model study of the sulphur cycle in the Arctic landfast ice zone. The quality of the text and figures is quite good. The authors have the interesting conclusion that sea ice sulphur cycle and ecosystems have considerable impacts on DMS production under the ice and should be considered in estimates of ocean DMS fluxes. Whereas I believe this conclusion is potentially supported by the scientific elements of the paper, the current presentation did not convince me.
1) I'm not sure that sulphur is conserved in the model. This question is central: if the authors suggest an enhancement of sulphur fluxes to the atmosphere, they must explain where this extra sulphur comes from. I don't see in the text or in the figures which reservoir is loosing sulphur in your model.

Sulfur is not conserved in our model. However, this is not essential to simulate the dynamics of sulfur species of our interest (i.e. DMSP and DMS), as sulfur (existing in elements and various compounds) is present abundantly in the ocean. Our focus is to quantify how much of this sulfur pool can be converted into the volatile compound DMS that can emit into the atmosphere. To the best of our knowledge, none of the marine sulfur cycle models that were developed to study the oceanic DMS production and emissions conserve sulfur either. Models can simulate DMSP and DMS adequately without conserving sulfur as long as the major production and removal processes for these sulfur species are incorporated (and they do not need to be balanced). In the model, the production of DMS originates from the production of DMSPp by primary producers. An enhancement of sulfur fluxes to the atmosphere is due to the incorporation of sea ice ecosystems that provide an additional source of DMSPp (and therefore DMSPd and DMS) for the underlying water column. We revised Sections 2.2 of the manuscript to clarify the non-conservation in the model.

2) I cannot understand the mechanisms in the sensitivity experiments from the text. Section 3.2.1, they authors explain they turn the sulfur cycle off. Which terms of the equations does that represent ? What happens with DMS in these experiments ? What is the chain of mechanisms leading to the decrease in under-ice DMSPd and DMS and decreasing the air-sea fluxes ?

In Section 3.2.1 of the sensitivity study, we turn off the sea ice component of the sulfur cycle (NoIceSul). This means that the sea ice sulfur cycle is completely absent in the model (i.e. zero concentration of DMSPd or DMS in the bottom ice). Consequently, all the sources and sinks for bottom-ice DMSPd and DMS presented in Equations A2 and A8 are zero (and therefore they are not computed) in this sensitivity run. Finally, this leads to a decrease in under-ice DMSPd and DMS because there is no supply of DMSP and DMS from the bottom ice into the underlying water column (represented now as "Release" in Fig. 1). We revised Section 3.2.1 to provide more detailed explanations for this mechanism. Furthermore, we created a figure showing the model schematics for the three different runs (i.e. Standard, NoIceSul, NoIceBgc), which is now added to the supplementary material (Fig. S2).

This is my key criticism. If the authors can at least highlight how sulphur is conserved among the different sulphur forms and, more importantly, explain more in depth the mechanisms in their sensitivity experiments, this can make a good paper.

— A few more detailed comments - Model description is not convincing. This could probably be fixed by better explanations (not more of them). 1) Physical and ice algal components models come from a paper under review. Next time the authors should consider to attach the companion paper.

We agree that the physical and ecological components of the model may not have been described adequately in our manuscript. We revised Section 2.1 to provide a better explanation for physical and ecological components of the model. We will certainly consider attaching the companion paper in future submission.

2) I doubt of sulphur conservation. I don't clearly see how sulphur can be conserved now. It should be visible from the equations. For instance, I don't see where the losses of DMSPp in sea ice go (eq A1). Make sure sulphur conservation is obvious from the few evolution equations.

We have addressed the concern about sulfur conservation in the first response above. DMSPp in sea ice is simulated diagnostically; it is defined as the product of ice algal biomass and the DMSP cell quota. Therefore, the temporal evolution of DMSPp in sea ice is proportional to that of ice algal biomass.

3) Some physical terms (notably sea ice growth and melt for the first ocean layer) are completely absent from the equations, which is surprising, because these are leading- order terms for most other biogeochemical compounds.

We agree that it is not clear from our manuscript alone to comprehend how some of the physical terms are defined in the model. Because the physical component of the model is derived from a model used in previous studies and is defined therein, we did not show their equations. We revised Appendix A of our manuscript to provide appropriate references for the definitions of physical terms used in the sulfur cycle model.

4) Is there any good reason not to use a standard formulation of the DMS air-sea flux? In your section 3.2.3, they are just proportional to DMS_water. As far as I know, this is not in line with classical air-sea flux formulations. It would not be hard to introduce solubility and pDMS in the atmosphere.

We agree with the referee that, for air-sea flux calculation, it would be more appropriate to account for the atmospheric DMS with solubility. However, it is a common practice to neglect the atmospheric DMS in the flux calculation based on the assumption that the oceanic DMS often exceeds the atmospheric DMS by orders of magnitude. Furthermore, incorporating the atmospheric DMS would require either simulation of atmospheric DMS or observed atmospheric DMS data set, but none of these are available. For these reasons, we removed the atmospheric DMS and solubility from the flux equation in this study. We revised the manuscript to clarify these points (Section 3.2.3).

- Sensitivity experiments are difficult to understand. I have trouble to distinguish between "sea ice ecosystem" and "sea ice sulphur cycle" sensitivity experiments, because the sulphur cycle is partly controlled by sea ice algae. So the authors should clearly tell which terms are involved and, most importantly explain the mechanisms involved.

We agree with the referee that the distinction between sea ice ecosystem and sea ice sulfur cycle could be clearer with better explanations. We revised Section 3.2 of the manuscript accordingly.

- In figures, the authors often compare model concentrations in μmol/L to observed ones. This could introduce a source of bias is the depth of the extracted core section does not match the 3 cm of the

model. I would suggest to rescale observations to 3 cm if possible.

The observed bottom ice DMSP and DMS concentrations in umol/L represent the bottom 3 cm of the extracted ice core, therefore there is no need to rescale.

- Some aspects of the intro (links between DMS and cloud nucleation) may not be in phase with literature. I felt the role of DMS was a little bit overstated. Line 10 of page 2, the authors point DMS as the driver of arctic clouds backed with one unique citation. I was surprised by this statement. I had a little trip in the literature, and discovered that this should probably be nuanced. Tjernström et al (ACP 2014) mention page 5 that they are indeed looking for a missing source of aerosols in the Arctic. Yet they did not found H2SO4 but rather organic molecules polymer saccharide molecules. Later on, page 2828, they explain that "This suggests a stronger possible link between marine biology, cloud prop- erties and cli- mate than provided by DMS alone (Leck and Bigg, 2007)". Tjernström et al. Atmospheric Chemistry and Physics, 14, 2823-2869, 2014. I'm sure the authors are aware of these works, and I would here just suggest to better explain the the links between DMS and Arctic clouds, even if it takes a few sentences.

We agree with the referee that the role of DMS was a little bit overstated in the sense that the role of other important marine biogenic precursors of CCN (such as microgels) should have been acknowledged. We revised the introduction section accordingly.

- The introduction felt generally a little bit "inbred", with lots of references coming from the own group of the authors. If there is no other choice, skip this comment.

We revised the introduction section to include few more references from the NETCARE field campaigns.

---

## Author Comment (AC2) · 3 Mar 2017

General comments: The study of Hayashida et al. introduces a new coupled sea ice-ocean ecosystem-sulfur cycle model. The purpose of the model is to determine the DMS,P production of Arctic sea ice microbial communities and its contribution to oceanic emissions of DMS in the Arctic. This contribution is important to assess since: -Oceanic DMS emissions could play a paramount role in aerosols formation in the rel- ative pristine atmosphere of the Arctic Ocean. Hence, DMS could be a key compound regulating the regional climate. -Sea ice microbial communities generally show very high production of DMS,P, with concentrations several orders of magnitude higher than typical oceanic concentrations. -The sea ice cover in the Arctic is changing rapidly in response to regional warming, generating potential important feedback loops. In my opinion, the general context of the work of Hayashida et al. is therefore highly relevant.

As correctly introduced by the authors, both the DMS,P production of sympagic com- munities and its contribution to oceanic emissions of DMS are both unfortunately cur- rently poorly constrained. Only a handful of studies have determined DMS,P (DMSPd, and DMSPp mainly, only one or two studies have reported DMS concentrations) con- centrations in Arctic sea ice, and only a couple have tackled production and removal processes and rates. Close to nothing is known about the transfer of DMS,P at the ice/ocean/atmosphere interfaces. In addition, sea ice typically shows a very high tem- poral and spatial variability making harsh any extrapolation of field studies that are limited in space and time. In this context, I am really happy to see the modelling effort developed by Hayashida et al. As detailed in my minor comments, it brings both inter- esting answers and questions to the sea ice DMS community and is therefore worth publishing.

As it is the first time such a perilous (given the complexity of the sea ice DMS cycle) enterprise is undertaken, there are however a lot of caveats and limitations that I think the authors could do a better job at presenting in the paper. This is in fact my major comment or request to the authors. The caveats and limitations are only partially tackled in the discussion and in the conclusion. I think they deserve their own section in the discussion part of the manuscript. I would like this section to: 1) Present the caveats and limitations (see my minor comments) of the model itself, in particular the fact that brine dynamics are neglected. 2) Present recommendations for observational studies, something like "what modellers need from field studies".

We thank the referee for the thoughtful suggestion. We created a new section on caveats and limitations (i.e. Section 3.3: Limitations of the present study). As you will see in our responses below, some of the questions/suggestions by the referee are answered in this new section.

This section could also tackle the second important weakness of the paper, the fact that the model is validated by only one time series study that does not even cover all the outputs of the model. Is there really not any other data available in the literature that you could have use to validate the model or at least to give an idea to the reader on the applicability of your model to other locations/ice conditions in the Arctic? The limited time frame of the Galindo et al data set which partially miss two important phases of the cycle presented by the model does not help to build confidence. I think that throughout the manuscript you could better put your results and parameters in perspective with the literature, even if you have to use the more abundant Antarctic sea ice DMS literature. I know that most of the drivers are significantly different in the Antarctic, but some basic physical and biological concepts are the same, especially if you look at fast ice studies with no surface flooding. It is also really important in this section, or in other sections of the paper, to better convince the reader that your model is applicable to other parts of the Arctic and not only to the study site of Galindo et al. A way to do this would be to better describe the study site and conditions of Galindo et al and to show that they correspond to other sites/and conditions in the Arctic.

We thank the referee for providing constructive feedback on the weakness of our manuscript. We revised the manuscript to provide more comparions of our results with other studies. We also created a new section called "Study site" to provide a general information of the study site.

One final major concern that I have reading the paper is the use of the term flushing throughout the manuscript. It is never described in the manuscript and I am having trouble knowing what the authors really refer to. Flooding is a very specific process defined as: "Flushing refers to the washing out of salty brine by relatively fresh surface melt water that percolates into the pore space during summer". I have the feeling that flushing is used as another term for "release" by the authors and could in fact include other form of material release from sea ice (e.g. brine drainage, melting of bottom ice). Could you please better define this in the manuscript and replace by another term if necessary? In addition to these general comments, I also have additional minor comments and suggestions listed here after:

As pointed out by the referee, we used the term "flushing" to include other form of release from sea ice (i.e. basal melting). In our model, two types of release are paramterized: 1) flushing due to the drainage of meltwater from melt ponds, as well as surface and interior ice; and 2) sloughing due to basal melting. We revised the manuscript throughoutly to clear the confusion.

Specific comments:

  Abstract

Page 1: Line 5: Please use the plural form: The result"s" of the 1-D model "were" compared. Also consider using plural for DMS emis- sion"s". Line 6: Please use the plural form: our result"s" reproduced.

Revised as suggested.

Line 7: Flushing is a very distinct process in sea ice thermodynamics. Please consider using the word "release" more generic than flushing, unless you are exclusively talking about flushing. This comment actually applies to the whole document.

Revised as suggested.

Line 9: "Processes that dominated the budgets of bottom-and under-ice. . ." This sentence is a little bit heavy, consider rephrasing.

Revised as suggested.

Line 15: "would be better constrained by new observations". The whole sentence sounds like a recommendation, consider using "should" instead of "would".

Revised as suggested.

âA˘ c´ Introduction

Page 1: Line 18: "is a volatile biogenic trace gas" A gas is always volatile, please use either volatile or gas in the sentence.

Revised as "volatile biogenic compound".

Line 19: "planktonic and microbial" Microbe relates to the size of organisms, and so includes some planktonic organisms. The use of the two words is a little bit confusing here.

Revised as "microbial".

Line 20: "Oceanic DMS emissions also play. . ." I would say "can play" especially considering your next paragraph about the controversy on the global scale effect of DMS.

Revised as suggested.

Page 2: Line 7,8: This sentence is very long, consider splitting it in two.

Revised as suggested.

Line 8: "increase and peak" I would say: "were observed to increase and peak".

Revised as suggested.

Line 16: ""high concentrations at the bottom" I would say "in the bottom".

Revised as suggested.

Line 19: I would stick only to Arctic studies here, so please remove Kirst et al., 1991 from the reference list, or clearly indicate that the study of Kirst et al. 1991 covers the Antarctic.

Removed as suggested.

Line 26: I would like to see one or two sentences saying what the outcome of the model study of Elliott

et al. was and how your study differs from their approach. I think it is important to frame your study.

Revised as suggested.

Line 27: Please consider starting the paragraph with "In this study, we test the hypothesis that. . . by developing a sulfur cycle. . .".

Revised as suggested.

Line 30: "The rest of this study. . ." I don't think this part is necessary unless required by the journal.

Removed as suggested.

  ć Model description and experimental design

Page 3: Line 14: "diffusive exchange of nutrients at the ice-water interface". Does (and if yes how) the model consider the role of brine convection in the exchange of nutrients at the ice-water interface? Several authors and Antarctic based models have highlighted the importance of this process, see Vancoppenolle et al. 2010 for instance ("Modeling brine and nutrient dynamics in Antarctic sea ice: the case of dissolved silica").

Our model does not consider the role of brine convection (via brine drainage) in the exchange of nutrients at the ice-water interface, although it is indirectly included by having a continuous diffusive exchange (e.g. Lavoie et al. 2005). We agree with the referee that including the effects of brine dynamics on nutrient dynamics would result in more realistic representation (Vancoppenolle et al. 2010). We revised Sec. 3.3. of the manuscript to make a note on this process for future model development.

Lavoie et al. 2005: Modeling ice algal growth and decline in a seasonally ice-covered region of the Arctic (Resolute Passage, Canadian Archipelago).

Line 23: Please add "emphasized that "the" sulfur cycle represented. . .".

Revised as suggested.

Line 32: It should be clearly stated here that cell lysis and exudation rates are taken from limited water column measurements and not sea ice/brine measurements.

Revised as suggested (not only cell lysis and exudation, but most of parameters for the simulated processes come from limited water column measurements).

Page 4: Line 4 to 7: This sentence is hard to understand, please rephrase or make 2 separated sentences. Same comment as in the abstract regarding the term "Flushing".

Revised as suggested.

Line 10: Good, I like to read this, but not only for S but also for nutrients, see my previous comment.

We revised Sec. 3.3. of the manuscript to make a note on the role of brine convection on nutrient

dynamics for future model development.

Line 14, 15: Ignoring sloppy feeding is unfortunately a weakness of the paper given the relative importance of grazing on bottom ice communities. Same for the DMSO reduction to DMS.

Although sloppy feeding was thought to be an important process for DMS production in the previous sea ice DMS modelling paper (Elliott et al. 2012), there was no indication for strong contribution from sloppy feeding during the Arctic-ICE 2010 study (Galindo et al. 2014). Therefore, we neglected this process. However, sloppy feeding in Elliott et al. (2012) and exudation in our study are parameterized similarly (i.e. both processes are linearly proportional to the modelled ice algal growth rate), therefore provide similar outcome. We revised the manuscript to discuss this in Sec. 3.3.

As for the DMSO reduction to DMS, we agree with the referee that this process can be an important pathway for DMS production in sea ice, although, to the best of our knowledge, there is only one study to support this (i.e. Asher et al. 2011: High concentrations and turnover rates of DMS, DMSP and DMSO in Antarctic sea ice). However, we had to neglect this process as there is no observational constraints on the rates for this process. We revised the manuscript to discuss this in Sec. 3.3.

Model setup: perhaps some general information about the Arctic-ICE Resolute passage sampling area could be provided to the reader here. Typ- ical ice thickness developing, type of ice (thickness, texture), typical date of formation and retreat, ice deformation, typical ice and weather regimes. . . I think that somewhere you need to convince the reader that your site is somewhat representative of a more general situation e.g. Arctic coastal fast ice.

We created a new section (Section 2.3: Study site) to provide a general information on the study site.

Line 27: What is the total water depth at the Arctic-ICE sampling site?

The depth of the water column was 141 m (Galindo et al. 2014). We revised Section 2.4 to indicate provide this information.

Line 31: Was a met tower deployed at the sampling site? If yes, why not using data from the tower? Or at least compare the two data sets.

A met tower was deployed within 500 m of the study site. However, only air temperature and irradiance were measured at this site (see Mundy et al. 2014). Because our model requires other meteorological variables for forcing (e.g. wind speed/direction, relative humidity, pressure), we used the Resolute airport data for consistency among these variables (i.e. they were all measured at the same location). In Supplementary material, We have added a new figure showing the time series of air temperature observed at Resolute airport, which are similar to the met tower time series. We revised Section 2.4 to briefly describe this and made a reference to the figure in the supplementary material.

Line 32: I would like to see in addition to Dukhovskoy et al. 2016 a more general reference for the NEMO-LIM2 model.

Both a general reference and specific details of NEMO-LIM2 used in Dukhovskoy et al. (2016) are provided in their Appendix (A2), therefore we believe it is sufficient to provide just the reference to

Dukhovskoy et al. (2016). We revised Section 2.4 to clarify that what a NEMO-LIM2 is (a coupled 3-D regional sea ice-ocean circulation model).

Line 34: Where the initial ice thickness, snow thickness, and melt-pond depths set based on field observations? This is not clear.

Initial snow and melt pond depths and ice thickness set respectively to 5, 0, and 55 cm, result in simulations of these variables in good agreement with the measurements from the Arctic-ICE 2010 field campaign. We revised Sec. 2.4 to clarify this justification.

Page 5: Line 1: "to match with observations" To match with observations of the real depth of the skeletal layer, or to match with the vertical sampling resolution used by Galindo et al.? I think this should be clarified.

Revised as suggested (To match with the vertical sampling resolution used by Galindo et al.).

Line 2: It is not clear from your writing if the initial biomass of ice algae was set based on the measurements of Mundy et al. (2014) as for the nutrients. Same comment for ammonium and particulate silica.

The initial biomass of ice algae is not based on Mundy et al. (2014), but set to a value that resulted in simulating reasonable ice algal biomass during the Arctic-ICE 2010 campaign. Although this value may seem high considering what is expected for Chl a in the water column, previous studies indicate a wide range of Chl a in young ice (0.3-26.8 ug L-1) that is often higher than Chl a in the water column (e.g. Garrison et al. 1983). For ammonium and particulate silica, values of 0.01 are set assuming they were small during the winter. We revised the manuscript to provide this information above.

Garrison et al. 1983: A physical mechanism for establishing algal populations in frazil ice.

Line 7: This is a strong assumption given that your initial chl a is set to 3.5 µg.L-1. In oceanic waters, for such chla concentrations you already have non negligible DMS and DMSPd concentrations. And it is definitely the case for sea ice as well, even in diatom dominated communities (see several Antarctic studies for instance, e.g. Carnat et al., 2014 "Physical and biological controls on DMS,P dynamics in ice-shelf influenced fast ice during a winter-spring and a spring-summer transitions" JGR.). How do you justify this?

In the water column, the initial biomass for small and large phytoplankton is set to a small value (0.01 mmol N m-3; ca. 0.035 ug Chl a L-1), therefore the initial DMSP and DMS concentrations (0.1 nmol L-1) set in the water column are also small, which is reasonable. For sea ice, given the initial value of 3.5 ug Chl a L-1 and the DMSP-to-Chl a ratio of 9.5, the initial DMSPp concentration in the bottom ice is about 33 nmol L-1. Considering that the model simulation starts from February 1 (i.e. middle of Arctic winter, biological proccesses are still low), the initial DMSPd and DMS concentrations in the bottom ice (0.1 nmol L-1) set in the model are reasonable. It is difficult to compare our initial values with the measurements reported in Carnat et al. (2014) because the environmental conditions are very

different; our initial values are representative for early-winter with growing ice (from 55 cm), while the first sampling in Carnat et al. (2014) was done in late-winter with much thickner ice (147 cm; Station 4).

Line 14: Are these "field measurements in Arctic waters" referenced somewhere in the manuscript?

They are referenced in the Appendix. We revised the manuscript to make a reference to the Appendix.

Line 15: "Emissions"s".

Revised as suggested.

âA˘c´ Results and discussions

Page 5: Line 30: How did you deal with melting and refreezing of the snow pack (typical in early summer in the Arctic) and the formation of superimposed ice? Was superimposed ice detected in the field samples? Was superimposed ice considered as snow or ice in the obs?

Our sea ice thermodynamic model does not simulate refreezing of snow pack and the formation of superimposed ice, as they require very complex parameterizations. We agree with the referee that superimposed ice was detected at the end of spring during the observations. Looking at the color, we were able to differentiate sea ice (blue) and superimposed ice (grey-white). We have the measurements for that if necessary. In the observed time series (Fig. 2), the superimposed ice was included in the ice thickness values. We added this unresolved process to Sec. 3.3 of the manuscript.

Page 6: Line 9: Yes, and this is why it is important to give general information about the sampling site to the reader. Is this assumption based on observations during the 2010 campaign or during other years? Do you have a reference for this?

This assumption was not based on the observations during the 2010 campaign. After a further literature review, we find that mechanical processes are not as important as thermodynamic processes in the region (e.g. Flato & Brown, 1996). We removed the last sentence in Section 3.1.1.

Flato & Brown (1996): Variability and climate sensitivity of landfast Arctic sea ice.

Line 24: What about the increase in observed under-ice chla in mid-May and the localized increase in early-June? Could they be the sign of brine drainage not simulated by the model? I think the hypothesis has to be written somewhere. You see the same trend in DMSPp and to a lesser extent in DMSPd.

We agree with the referee that the increase in observed under-ice chl a, DMSPp, and DMSPd can be due to brine drainage lacking in our model. In fact, Galindo et al. (2014) attributed the loss of ice algae and ice-DMSPp prior to the snowmelt period to the effect of brine drainage. However, it is unlikely that brine drainage played a role in early June as bulk salinity measurements presented in Galindo et al. 2014 showed little variation during this period. We attribute this mismatch in early June as underestimates in simulated flushing. We revised Section 3.1.2 and 3.1.3 to provide these explanations.

Line 27: It would be nice to indicate here the fixed DMSP:chla ratio used and not only refer to the

equations.

Revised to indicate the ratios and made reference to Table 2.

Line 28: How strong was the relationship? Please indicate this.

$R^2 = 0.9$, revised to indicate this value.

Line 32: How do you explain the spatial variability? Spacing of brine channels? Patchiness of ice algae in bottom ice? Hints should be given here.

Patchiness of ice algae, as the spatial variability here reflects the sampling done over various snow cover sites (high, medium, and low). We revised Section 3.1.3 to include this possible explanation.

Page 7: Line 9 and 10: "To the release of ice algae from bottom ice". Through which process? Bottom sea ice melting? Hanging algae that just drop in the water? Brine drainage? Flushing?

Brine drainage. We revised the manuscript accordingly.

Line 13 and 14: The DMSPp sinking part does not convince me at all. What is the argument to support this? Just that the DMSPp peak rapidly disappears during the next sampling event? Do you have DMSPp measurements in deeper ocean layers during the same sampling event? It could be also that most of the DMSPp is rapidly converted to DMSPd and DMS. You do not have observations of DMS, so you cannot rule that second option out. I think that you should mention both options and not only focus on the sinking of DMPSp.

We agree with the referee that both explanations for rapid loss of DMSPp in the water column are plausible and thus both should be mentioned. We have observations for deeper depths (up to 50 m presented in Fig. 8 of Galindo et al. 2014, as well as unpublished result for depth at 130 m) that shows slight increase in Chl a and DMSPp (and larger increase in DMSPd) at deeper depths on the following sampling days. We revised the manuscript to provide these explanations in the fourth paragraph of Section 3.1.3.

Line 23 to 25: The explanation given for the mismatch between the three first observations and the model simulation in DMSPd does not convince me at all for the moment. We need to know at which snow depth day 1 was sampled and what the effect could be on DMSPd. Why would spatial variability be higher during day 2 than during any other sampling day? Is the fact that all the three observed variables (DMSPp, DMSPd, chla) were significantly higher than model predictions during day 3 not at sign that the model is missing something? What about other ancillary parameters observed during the 3 first sampling days? (community composition for instance?, stress on the algae?).

On day 1, sampling was done over a medium snow cover site. We do not know the cause of the high spatial variability on day 2. There was no significant change in community composition throughout the sampling period (Galindo et al. 2014). Considering that brine drainage had occurred during these sampling days, we believe that the stress on the algae might be a possible cause. We revised Section 3.1.3 to provide better explanation.

Line 31: "Degradation of DMSPp to DMSPd" that you disregarded before, saying that DMSPp was

sinking. . .see line 13 and 14.

Same comment as the response to Lines 13 and 14.

Page 8: Line 3: "upper 10 m" perhaps add "of the water column".

Revised as "upper 10 m average of the water column".

Line 7: Considering the comment you make Line 28-30 of page 11 on the cell lysis and artificial DMSP release (and hence DMSP conversion to DMS) on melting of ice samples, how can we trust the 2000 nmol.L-1 value given in Levasseur 2013. How was the sea ice DMS measured in that study?

We agree with the referee that the measurements of bottom-ice DMS by melting will likely result in overestimates (as was done in Levasseur 2013). However, we also do not know well about the magnitude of this overestimate. Furthermore, considering that this is the only published ice-DMS value for Resolute Passage, we feel that it should be compared with our simulated peak.

Section 3.1.4.: I know there is no sea ice field observations to validate these various production and removal rates but it would be nice to at least compare them to oceanic field observations so that the reader can evaluate if the numbers make sense.

We did a further literature review and noticed that Asher et al. (2011) report observed rates of DMS consumption and DMSP cleavage both in sea ice brine and ice-covered seawater samples. We revised Sec. 3.1.4 to add these rates for evaluation of our simulated results. In short, they are very comparable. Furthermore, we provide a brief discussion on the similarity/difference between our simulated rates in bottom-ice/under-ice environment and the observed rates in open-water environment.

Asher et al. (2011): High concentrations and turnover rates of DMS, DMSP and DMSO in Antarctic sea ice.

Page 9: Line 1: "due to low zooplankton biomass during the melt period" Is this supported by any kind of observation?

No measurements of zooplankton biomass were conducted during this period of the Arctic-ICE 2010 study, therefore the model result could not be observationally assessed. A previous study suggest high interannual variability in zooplankton biomass in Resolute Passage (Michel et al. 2006). We revised Section 3.1.4 to include this information.

Michel et al. (2006): Variability in oceanographic and ecological processes in the Canadian Arctic Archipelago.

Line 21-22: "In mid-June, the simulated DMS. . ." This sentence is hard to understand, please rephrase or cut in two separated sen- tences.

Revised as suggested.

Line 25-30: This whole part is a little bit hard to follow. It is hard to know when you are talking about pre bloom conditions, early bloom conditions, peak bloom conditions...I would suggest rewriting like this (line 25): "On the other hand, during the initiation of the under-ice bloom, the simulated DMS production rates by flushing were comparable to the rates of other simulated processes associated with the bloom, reaching a maximum value (>5 nmolL.d) in early June. "

Actually this sentence (line 25) and the previous sentence (line 23-24) are talking about the same bloom condition (prior to the under-ice bloom = early June). We agree with the referee that the way it was written in the manuscript was somewhat confusing. We merged these two sentences into one and shortened.

Page 10: Line 1-2: What about the upward vertical migration of DMS within the brine channels? It is very likely that in an open system such as warm interconnected brine channels, gaseous compounds can migrate upwards due to buoyancy if present under the form of bubbles. Would DMS form bubbles at some point? See Zhou et al., 2013 "Physical and biogeochemical properties in landfast sea ice (Barrow, Alaska): Insights on brine and gas dynamics across seasons", or Moreau et al. 2014 "Modelling argon dynamics in first-year sea ice". Did you consider this? If not, why? I think this needs some clarification.

We did not consider the bubble nucleation and its upward migration within the brine channels in our model at this point, as we felt that implementation of this process into the model requires observational constraints (such as done for Argon). To the best of our knowledge, we do not know of any published work on this bubble process for DMS. We revised Section 3.3 to include a discussion on this process.

Line 23: The use of the word "significantly" suggests that statistical tests were conducted. Is it the case? And if yes, please indicate the level of signifi- cance. If no, please use another term.

Revised as suggested (replaced "significantly" with "much").

Line 24-25: Please rephrase: "In comparison to the NoIceSul run, the NoIceBgc run simulated under-ice DMSPd and DMS concen- trations that were higher in late June onward."

Revised as suggested.

Line 28: "Heightened". . . "higher"?

 Revised as suggested.

Line 27-28: This needs further explanation. How does the absence of ice algae increase the nutrients available to under-ice algae? It is not clear from you text how they get access to the same nutrient pool? Is it the fact that the nutrients not consumed in bot- tom ice are "flushed" to the under-ice water ? Or the fact that the under-ice nutrient pool is somehow consumed by bottom ice algae? If yes, how? Also this whole section lacks comparisons with observations of nutrient/algae dynamics from the literature. It would also be nice here to remind here that the effect of brine convection on nutrient dynamics (see

Vancoppenolle work) was not taken into account in the model.

In our model, nutrients in the bottom ice are exchanged with the nutrients in the uppermost layer of the water column (i.e. under-ice nutrients) through diffusive mixing, which is parameterized similarly to Lavoie et al. (2005). Detail of the parameterization is provided in Mortenson et al. (submitted.). In short, the exchange rate depends both on the horizontal ocean currents (which determine the strength of turbulence through shear) and the gradient of nutrient concentrations between the bottom ice and the uppermost water column. In the presence of ice algae, nutrients are consumed in the bottom ice and replenished from the uppermost water column via diffusive mixing, which indirectly decreases the under-ice nutrients. We did not provide comparisons of nutrient/algae dynamics from the literature because they are done in Mortenson et al. (submitted.). We revised Section 3.2.1 to add the explanation for nutrient increase in the absence of ice algae and made a note about brine convection.

Line 34: This sentence is too long and complex, please split in two or simplify.

Revised as suggested.

Page 11: Line 4: of "the" sea ice ecosystem. Line 10: Remove one of the "the".

Revised as suggested.

Again, you need to explain to the reader how the dynamics of the bloom were precisely affected. Whole section: What about the self-shading effect? Removing bottom ice algae should increase the amount of light available to under-ice algae. Did you consider this? Line 10: Influencing the dynamic of the bloom. . .through the control exerted on nutri- ents? How?

We considered the shading effect of ice algae, which should delay the onset of the under-ice bloom due to reduction in light available under the ice. Thus, the shading effect has an opposite effect from the seeding effect. The results of the standard and the NoIceBgc runs suggest that the latter had a stronger impact on the bloom dynamics as the exclusion of ice algae did not result in an earlier onset. (Fig. 6a). We revised Sec. 3.2.1 to provide full explanations.

Line 22-25: This sentence is too long, please split in two.

Revised as suggested.

Line 25-30 and line 28-30: You should clearly state here which study were conducted in the Arctic and which were conducted in Antarctica. Also, you need to discuss major algal groups in the assemblages sampled by the different studies. Line 28-30: Right, I agree with this. Cell rupture on melting of the samples will cause artificial release of DMSPp and trans- formation into DMSPd and DMS in contact with free or cell bound enzymes. But then, how can you trust the DMSPp and DMSPd measurements of Galindo et al that you use to validate the model?? Furthermore, how can you trust the DMS measurement from Levasseur 2013? All these measurements were made by melting sea ice samples. There is at least one study in the literature that reports DMSP:chla ratios using other techniques than melting, you should refer to it. For example, Carnat et al. 2014 "Phys- ical and biological controls on DMS,P dynamics in ice-shelf influenced fast ice during a winter-spring and a spring-summer

transitions" JGR.) gives DMSPt:chla ratios for bottom ice diatom dominated communities. I think the best way to present this whole DMSP:chla issue is to draw a quick table with a few references from the literature in- dicating DMSP(p or t):chla ratios measured, community compositions, arctic/Antarctic, melting or other extraction technique. This table could go in the supplementary but would be a very nice addition to the paper.

We appreciate the referee's suggestion to create a table to facilitate this DMSP:Chl a ratio overview. We created and added the table in the manuscript (Table 2). As our focus is the ratio representative for bottom-ice community, the table only contains values of this ratio for diatom-dominated community. We also realized that the two references used in the manuscript (i.e. Uzuka 2003 and Kirst et al. 1991) report the total DMSP-to-Chl a ratio, rather than the particulate DMSP-to-Chl a ratio. What we need for our model is the latter, and Galindo et al. 2014 and Levasseur et al. 1994 are the only two studies that report this ratio. Therefore, despite the potential error in DMSPp measurements due to melting sea ice samples, we argue that the DMSPp-to-Chl a ratio reported in Galindo et al. 2014 would be the best starting point for modelling exercise. We revised Section 3.2.2 to discuss these points.

Page 12: Line 5-6: It would be nice here to restate quickly which parameters then control the temporal patterns of DMS concentrations.

Revised as suggested. The temporal patterns of DMS concentrations are more or less invariant, as they are controlled by the ecosystem dynamics, which are identical among these simulations.

Line 14-15-16: This is very nice to read.

Line 23-24: I am not sure to understand this correctly. Is this because DMSPd removed by bacterial consumption is not available anymore to free DMSPd lyase? Is there because part of the DMSPd pool used by bacteria is converted partially in other compounds than DMS?

Yes (to the latter question). Although we cannot explain the magnitude change (2% increase) in DMS, the net effect of increasing the DMSPd consumption rate constant on DMS was a decrease because the model considers that only a fraction (set to 0.2) of DMSPd consumed by bacteria is converted to DMS, while the remaining fraction is lost as a sink. We revised Section 3.2.2 to provide this explanation.

Line 25: Ok, but which other parameters would you recommend observational studies to target?

The bacterial DMSPd consumption rate constant is the only parameter of our sulfur cycle model that has influence on multiple sulfur cycle processes. Although this is beyond our scope, if we were to extend the parameter sensitivity study to ecosystem model parameters, two other paramters (namely cell lysis rate constant and active exudation fraction) would be worth investigating because they have influence on both ecological and sulfur processes. We revised the last paragraph of Section 3.2.2 to make a note on these parameters. The two most influential parameters we recommend for observational studies to target are mentioned in the previous paragraph, therefore we do not think that they need to be restated here.

Page 13: Line 15: How did you deal with fetch of small leads and even large leads com- pare to open ocean parametrizations that you use? Also, how did you deal with shear- driven and convection-driven

turbulence in the sea ice zone that have been shown to have a huge impact on k (see Loose et al., 2014; A parameter model of gas exchange for the seasonal sea ice zone.) I do not especially ask you to redo all the math with a new or corrected parameterization, but at least the fact that these important parameters were neglected should be mentioned.

Our gas transfer velocity parameterization is based on Nightingale et al. (2000), therefore does not take into account the effects of fetch-limitation and ice-associated shear- and convection-driven turbulence. We revised the first paragraph of Section 3.2.3 to make a note on this limitation and the presence of more appropriate parameterization (i.e. Loose et al. 2014).

Line 16: Again, was any met tower deployed on site?

Yes, but winds were not measured at this tower (see Mundy et al. 2014).

Line 29-30: Pandis and Russel 1994 are old papers, is there not more recent work on particle formation associated with DMS flux?

To the best of our knowledge, no.

Line 30-31: "saturation" is an odd term to use here. I think the concept could be better explained with one or two additional sentences.

Revised as suggested.

Page 14: Line 16: Occurrence instead of occurrance.

Revised as suggested.

Page 15: Line 1: A reference for the summertime flux for ice free waters in the Arc- tic is missing here. I don't know however if the journal allows ref in the conclusion.

These references are already provided in Sec. 3.2.3, hence we do not include them in Conclusions.

Line 2 to 4: I totally agree with this part but I believe it should be a distinct discus- sion section of the paper rather than a few lines in the conclusion as mentioned in the general comments.

As suggested, we created a new section (Section 3.3) to discuss this part.

Line 13: I am a little bit surprised that nowhere in your recommen- dations/conclusions you are talking about the need for additional DMS,DMSP, DMSO data to validate models. I am also surprised that you do not say a word about brine drainage, although I believe it is a key missing item in your paper.

Revised as suggested.

Figure 1: Same comment as in the abstract regarding the term "Flushing".

Replaced the term "Flushing" with "Release".

Figure 3: The use of the different vertical scales for DMSPp, DMSPd, and DMS is slightly confusing. Is there a way to use similar vertical scales for the three compounds (by adding breaks or log scale?) so that they could be better compared?

We tried to incorporate this suggestion, but found that using similar vertical scales made it difficult to show the features of individual compounds in the time series. Therefore, we kept Fig. 3 as it is. For clarification, we made a note in Fig. 3 that the vertical scales are different in each graph.

Figure 4: Please check the legend. In the version that I have, the dashed lines do not show up. Also, in all graphs, please use parenthesis instead of the concentration symbol in the axis titles. Figure 5: Same comment as for Figure 5, dashed lines do not show up in the legend box.

Revised as suggested.

Figure 8: Perhaps indicate in the figure caption the distance between the sampling site and the Resolute Airport.

Revised as suggested.

Other figures are fine.

---

## Author Comment (AC3) · 3 Mar 2017

Please find attached our revised manuscript.

Please also note the supplement to this comment:
http://www.biogeosciences-discuss.net/bg-2016-399/bg-2016-399-AC3-supplement.pdf

---

## Author Comment (AC5) · 3 Mar 2017

March 3, 2017

[Figure]

Figure S1: Time series of daily mean surface 2-m air temperature observed at Resolute airport during 2010. The upper and lower vertical bars associated with the daily mean values represent the daily maximum and minimum values, respectively.

Figure S2: Comparison of model schematic among (a) the standard run, (b) the NoIceSul run, and (c) the NoIceBgc run.